# Random Weight Factorization improves the training of Continuous Neural Representations

## Abstract

Continuous neural representations have recently emerged as a powerful and flexible alternative to classical discretized representations of signals. However, training them to capture fine details in multi-scale signals is difficult and computationally expensive. Here we propose *random weight factorization* as a simple drop-in replacement for parameterizing and initializing conventional linear layers in coordinate-based multi-layer perceptrons (MLPs) that significantly accelerates and improves their training. We show how this factorization alters the underlying loss landscape and effectively enables each neuron in the network to learn using its own self-adaptive learning rate. This not only helps with mitigating spectral bias, but also allows networks to quickly recover from poor initializations and reach better local minima. We demonstrate how *random weight factorization* can be leveraged to improve the training of neural representations on a variety of tasks, including image regression, shape representation, computed tomography, inverse rendering, solving partial differential equations, and learning operators between function spaces.

## 1 Introduction

Some of the recent advances in machine learning can be attributed to new developments in the design of *continuous neural representations*, which employ coordinate-based multi-layer perceptrons (MLPs) to parameterize discrete signals (e.g. images, videos, point clouds) across space and time. Such parameterizations are appealing because they are differentiable and much more memory efficient than grid-sampled representations, naturally allowing smooth interpolations to unseen input coordinates. As such, they have achieved widespread success in a variety of computer vision and graphics tasks, including image representation (Stanley, 2007; Nguyen et al., 2015), shape representation (Chen & Zhang, 2019; Park et al., 2019; Genova et al., 2019; 2020), view synthesis (Sitzmann et al., 2019; Saito et al., 2019; Mildenhall et al., 2020; Niemeyer et al., 2020), texture generation (Oechsle et al., 2019; Henzler et al., 2020), etc. Coordinate-based MLPs have also been applied to scientific computing applications such as physics-informed neural networks (PINNs) for solving forward and inverse partial differential equations (PDEs) Raissi et al. (2019; 2020); Karniadakis et al. (2021), and Deep Operator networks (DeepONets) for learning operators between infinite-dimensional function spaces Lu et al. (2021); Wang et al. (2021e).

Despite their flexibility, it has been shown both empirically and theoretically that coordinate-based MLPs suffer from "spectral bias" (Rahaman et al., 2019; Cao et al., 2019; Xu et al., 2019). This manifests as a difficulty in learning the high frequency components and fine details of a target function. A popular method to resolve this issue is to embed input coordinates into a higher dimensional space, for example by using Fourier features before the MLP (Mildenhall et al., 2020; Tancik et al., 2020). Another widely used approach is the use of SIREN networks (Sitzmann et al., 2020), which employs MLPs with periodic activations to represent complex natural signals and their derivatives. One main limitation of these methods is that a number of associated hyper-parameters (e.g. scale factors) need to be carefully tuned in order to avoid catastrophic generalization/interpolation errors. Unfortunately, the selection of appropriate hyper-parameters typically requires some prior knowledge about the target signals, which may not be available in some applications.

More general approaches to improve the training and performance of MLPs involve different types of normalizations, such as Batch Normalization (Ioffe & Szegedy, 2015), Layer Normalization (Ba et al., 2016) and Weight Normalization (Salimans & Kingma, 2016). However, despite their remarkable success in deep learning benchmarks, these techniques are not widely used in MLP-based neural representations. Here we draw motivation from the work of (Salimans & Kingma, 2016; Wang et al., 2021a) and investigate a simple yet remarkably effective re-parameterization of weight vectors in MLP networks, coined as *random weight factorization*, which provides a generalization of Weight Normalization and demonstrates significant performance gains. Our main contributions are summarized as

- We show that *random weight factorization* alters the loss landscape of a neural representation in a way that can drastically reduce the distance between different parameter configurations, and effectively assigns a self-adaptive learning rate to each neuron in the network.

- We empirically illustrate that *random weight factorization* can effectively mitigate spectral bias, as well as enable coordinate-based MLP networks to escape from poor intializations and find better local minima.

- We demonstrate that *random weight factorization* can be used as a simple drop-in enhancement to conventional linear layers, and yield consistent and robust improvements across a wide range of tasks in computer vision, graphics and scientific computing.

## 2 WEIGHT FACTORIZATION

Let $x \in \mathbb{R}^d$ be the input, $g^{(0)}(x) = x$ and $d_0 = d$. We consider a standard multi-layer perceptron (MLP) $f_{\theta}(x)$ recursively defined by

$$f_{\theta}^{(l)}(x) = W^{(l)} \cdot g^{(l-1)}(x) + b^{(l)}, \quad g^{(l)}(x) = \sigma(f_{\theta}^{(l)}(x)), \quad l = 1, 2, \ldots, L, \qquad (2.1)$$

with a final layer

$$f_{\theta}(x) = W^{(L+1)} \cdot g^{(L)}(x) + b^{(L+1)}, \qquad (2.2)$$

where $W^{(l)} \in \mathbb{R}^{d_l \times d_{l-1}}$ is the weight matrix in $l$-th layer and $\sigma$ is an element-wise activation function. Here, $\theta = \left(W^{(1)}, b^{(1)}, \ldots, W^{(L+1)}, b^{(L+1)}\right)$ represents all trainable parameters in the network.

MLPs are commonly trained by minimizing an appropriate loss function $\mathcal{L}(\theta)$ via gradient descent. To improve convergence, we propose to factorize the weight parameters associated with each neuron in the network as follows

$$w^{(k,l)} = s^{(k,l)} \cdot v^{(k,l)}, \quad k = 1, 2, \ldots, d_l, \quad l = 1, 2, \ldots, L + 1, \qquad (2.3)$$

where $w^{(k,l)} \in \mathbb{R}^{d_{l-1}}$ is a weight vector representing the $k$-th row of the weight matrix $W^{(l)}$, $s^{(k,l)} \in \mathbb{R}$ is a trainable scale factor assigned to each individual neuron, and $v^{(k,l)} \in \mathbb{R}^{d_{l-1}}$. Consequently, the proposed weight factorization can be written by

$$W^{(l)} = \text{diag}(s^{(l)}) \cdot V^{(l)}, \quad l = 1, 2, \ldots, L + 1. \qquad (2.4)$$

with $s \in \mathbb{R}^{d_l}$.

### 2.1 A GEOMETRIC PERSPECTIVE

In this section, we provide a geometric motivation for the proposed weight factorization. To this end, we consider the simplest setting of a one-parameter loss function $\ell(w)$. For this case, the weight factorization is reduced to $w = s \cdot v$ with two scalars $s, v$. Note that for a given $w \neq 0$ there are infinitely many pairs $(s, v)$ such that $w = s \cdot v$. The set of such pairs forms a family of hyperbolas in the $sv$-plane (one for each choice of signs for both $s$ and $v$). As such, the loss function in the $sv$-plane is constant along these hyperbolas.

Figure 1 gives a visual illustration of the difference between the original loss landscape as a function of $w$ versus the loss landscape in the factorized $sv$-plane. In the left panel, we plot the original loss function as well as an initial parameter point, the local minimum, and

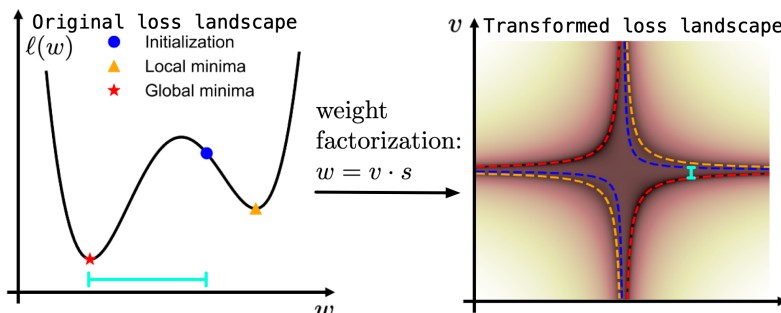

Figure 1: Weight factorization transforms loss landscapes and shortens the distance to minima.

the global minimum. The right panel shows how in the factorized parameter space, each of these three points corresponds to two hyperbolas in the $sv$-plane. Note how the distance between the initialization and the global minima is reduced from the top to the bottom panel upon an appropriate choice of factorization. The key observation is that the distance between factorizations representing the initial parameter and the global minimum become arbitrarily small in the $sv$-plane for larger values of $s$. Indeed, we can prove that this holds for any general loss function in arbitrary parameter dimensions (the proof is provided in Appendix A.1).

**Theorem 1.** *Suppose that $\mathcal{L}(\boldsymbol{\theta})$ is the associated loss function of a neural network defined in equation 2.1 and equation 2.2. For a given $\boldsymbol{\theta}$, we define $U_{\boldsymbol{\theta}}$ as the set containing all possible weight factorizations*

$$U_{\boldsymbol{\theta}} = \left\{ (\boldsymbol{s}^{(l)}, \boldsymbol{V}^{(l)})_{l=1}^{L+1} : \mathrm{diag}(\boldsymbol{s}^{(l)}) \cdot \boldsymbol{V}^{(l)} = \boldsymbol{W}^{(l)}, \quad l = 1, \dots, L+1 \right\}. \tag{2.5}$$

*Then for any $\boldsymbol{\theta}, \boldsymbol{\theta}'$, we have*

$$dist(U_{\boldsymbol{\theta}}, U_{\boldsymbol{\theta}'}) := \min_{\boldsymbol{x} \in U_{\boldsymbol{\theta}}, \boldsymbol{y} \in U_{\boldsymbol{\theta}'}} \|\boldsymbol{x} - \boldsymbol{y}\| = 0. \tag{2.6}$$

## 2.2 SELF-ADAPTIVE LEARNING RATE FOR EACH NEURON

A different way to examine the effect of the proposed weight factorization is by studying its associated gradient updates. Recall that a standard gradient descent update with a learning rate $\eta$ takes the form

$$\boldsymbol{w}_{n+1}^{(k,l)} = \boldsymbol{w}_n^{(k,l)} - \eta \frac{\partial \mathcal{L}}{\partial \boldsymbol{w}_n^{(k,l)}}. \tag{2.7}$$

The following theorem derives the corresponding gradient descent update expressed in the original parameter space for models using the proposed weight factorization.

**Theorem 2.** *Under the weight factorization of equation 2.3, the gradient descent update is given by*

$$\boldsymbol{w}_{n+1}^{(k,l)} = \boldsymbol{w}_n^{(k,l)} - \eta \left( [s_n^{(k,l)}]^2 + \|\boldsymbol{v}_n^{(k,l)}\|_2^2 \right) \frac{\partial \mathcal{L}}{\partial \boldsymbol{w}_n^{(k,l)}} + \mathcal{O}(\eta^2), \tag{2.8}$$

*for $l = 1, 2, \dots, L+1$ and $k = 1, 2, \dots, d_l$.*

The proof is provided in Appendix A.2. By comparing equation 2.7 and equation 2.8, we observe that the weight factorization $\boldsymbol{w} = s \cdot \boldsymbol{v}$ re-scales the learning rate of $\boldsymbol{w}$ by a factor of $(s^2 + \|\boldsymbol{v}\|_2^2)$. Since $s, \boldsymbol{v}$ are trainable parameters, this analysis suggests that this weight factorization effectively assigns a self-adaptive learning rate to each neuron in the network. Similar analysis has been explored in the context of deep linear networks , suggesting that the weight factorization can significantly accelerate the optimization . In the following sections, we will demonstrate that the proposed weight factorization (under an appropriate initialization of the scale factors), not only helps with mitigating spectral bias (Rahaman et al., 2019; Bietti & Mairal, 2019; Tancik et al., 2020; Wang et al., 2021c), but also allows networks to quickly move away from a poor initialization and reach better local minima faster.

### 2.3 RELATION TO EXISTING WORKS

The proposed weight factorization is largely motivated by *weight normalization* (Salimans & Kingma, 2016), which decouples the norm and the directions of the weights associated with each neuron as

$$\boldsymbol{w} = g \frac{\boldsymbol{v}}{\|\boldsymbol{v}\|}, \tag{2.9}$$

where $g = \|\boldsymbol{w}\|$, and gradient descent updates are applied directly to the new parameters $\boldsymbol{v}, g$. Indeed, this can be viewed as a special case of the proposed weight factorization by setting $\boldsymbol{s} = \|\boldsymbol{w}\|$ in equation 2.3. In contrast to weight normalization, our weight factorization scheme allows for more flexibility in the choice of the scale factors $\boldsymbol{s}$.

We note that SIREN networks (Sitzmann et al., 2020) also employ a special weight factorization for each hidden layer weight matrix,

$$\boldsymbol{W} = \omega_0 * \hat{\boldsymbol{W}}, \tag{2.10}$$

where the scale factor $\omega_0 \in \mathbb{R}$ is a user-defined hyper-parameter. Although the authors attribute the success of SIREN to the periodic activation functions in conjunction with a tailored initialization scheme, here we will demonstrate that the specific choice of $\omega_0$ is the most crucial element in SIREN's performance, see Appendix G, H for more details.

It is worth pointing out that the proposed weight factorization also bears some resemblance to the adaptive activation functions introduced in (Jagtap et al., 2020), which modifies the activation of each neuron by introducing an additional trainable parameter $\boldsymbol{a}$ as

$$\boldsymbol{g}^{(l)}(\boldsymbol{x}) = \sigma(\boldsymbol{a}\boldsymbol{f}^{(l)}(\boldsymbol{x})). \tag{2.11}$$

In practice, the scale factor is generally initialized as $\boldsymbol{a} = \boldsymbol{1}$, yielding a trivial weight factorization. As illustrated in the next section, this is fundamentally different from our approach as we initialize the scale factors $\boldsymbol{s}$ by a random distribution and re-parameterize the weight matrix accordingly. In Section 4 we demonstrate that, by initializing $\boldsymbol{s}$ using an appropriate distribution, we can consistently outperform both weight normalization, SIREN, and adaptive activations across a broad range of supervised and self-supervised learning tasks.

## 3 RANDOM WEIGHT FACTORIZATION IN PRACTICE

Here we illustrate the use of weight factorization through the lens of a simple regression task. Specifically, we consider a smooth scalar-valued function $f$ sampled from a Gaussian random field using a square exponential kernel with a length scale of $l = 0.02$. This generates a data-set of $N = 256$ observation pairs $\{x_i, f(x)_i\}_{i=1}^{N}$, where $\{x_i\}_{i=1}^{N}$ lie on a uniform grid in $[0, 1]$. The goal is to train a network $f_{\boldsymbol{\theta}}$ to learn $f$ by minimizing the mean square error loss $\mathcal{L}(\boldsymbol{\theta}) = 1/N \sum_{i=1}^{N} |f_{\boldsymbol{\theta}}(x_i) - f(x_i)|^2$.

The proposed random weight factorization is applied as follows. We first initialize the parameters of an MLP network via the Glorot scheme (Glorot & Bengio, 2010). Then, for every weight matrix $\boldsymbol{W}$, we proceed by initializing a scale vector $\exp(\boldsymbol{s})$ where $\boldsymbol{s}$ is sampled from a multivariate normal distribution $\mathcal{N}(\boldsymbol{\mu}, \sigma\mathrm{I})$. Then every weight matrix is factorized by the associated scale factor as $\boldsymbol{W} = \mathrm{diag}(\exp(\boldsymbol{s})) \cdot \boldsymbol{V}$ at initialization. Finally, We train this network by gradient descent on the new parameters $\boldsymbol{s}, \boldsymbol{V}$ directly. This procedure is summarized in Appendix B, along with a simple JAX Flax implementation (Heek et al., 2020) in Appendix C. The use of exponential parameterization is motivated by Weight Normalization (Salimans & Kingma, 2016) to strictly avoid zeros or very small values in the scale factors and allow them to span a wide range of different magnitudes. An ablation study in Appendix D further validates its benefit.

In Figure 2, we train networks (3 layers, 128 neurons per layer, ReLU activations) to learn the target function using: (a) a conventional MLP, (b) an MLP with adaptive activations (AA) (Jagtap et al., 2020), (c) an MLP with weight normalization (WN) (Salimans & Kingma, 2016), and (d) an MLP with the proposed random weight factorization scheme (RWF). Evidently, RWF yields the best predictive accuracy and loss convergence. Moreover, we plot the relative change of the weights in the original (unfactorized) parameter space during training in the bottom middle panel. We observe that

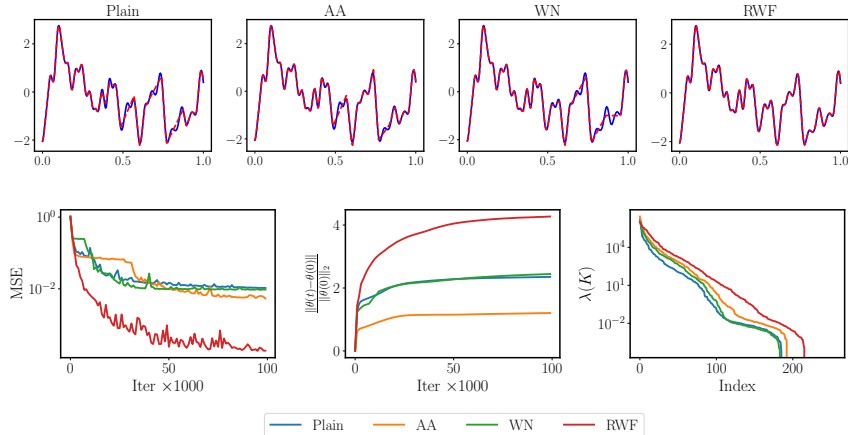

Figure 2: *1D regression: Top:* Model predictions using different parameterizations. Plain: Standard MLP; AA: adaptive activation; WN: weight normalization; RWF: random weight factorization. *Bottom left:* Mean square error (MSE) during training. *Bottom Middle:* Relative change of weights during training. The comparison is performed in the original parameter space. *Bottom right:* Eigenvalues (descending order) of the empirical NTK at the end of training.

RWF leads to the largest weight change during training, thereby enabling the network to find better local minima further away from its initialization. To further emphasize the benefit of weight factorization, we compute the eigenvalues of the resulting empirical Neural Tangent Kernel (NTK) (Jacot et al., 2018) $\boldsymbol{K_\theta} = \left\langle \frac{\partial f_\theta}{\partial \boldsymbol{\theta}}(x_i), \frac{\partial f_\theta}{\partial \boldsymbol{\theta}}(x_j) \right\rangle_{ij}$, at the last step of training and visualize them in the bottom right panel. As demonstrated in Tancik et al. (2020) and Wang et al. (2021c), the eigenvalues of the Neural Tangent Kernel (NTK) characterize the convergence rate of the corresponding kernel eigenvectors. Specifically, components of the target function that are aligned with kernel eigenvectors corresponding to larger eigenvalues will be learned faster. Therefore, the rapid decay of the NTK eigenvalues implies extremely slow convergence to high frequency components of the target function, a pathology commonly referred to as "spectral bias". Notice how RWF exhibits a flatter NTK spectrum and slower eigenvalue decay than the other methods, indicating better-conditioned training dynamics and less severe spectral bias. To explore the robustness of the proposed RWF, we conduct a systematic study on the effect of $\mu$ and $\sigma$ in the initialization of the scale factor $\boldsymbol{s}$. The results suggest that the choice of $\mu, \sigma$ plays an important role. Specifically, too small $\mu, \sigma$ values may lead to performance that is similar to a conventional MLP, while setting $\mu, \sigma$ too large can result in an unstable training process. We empirically find that $\mu = 1, \sigma = 0.1$ consistently improves the loss convergence and model accuracy for the vast majority of tasks considered in this work. Additional details are presented in Appendix D.

# 4    EXPERIMENTS

In this section, we demonstrate the effectiveness and robustness of *random weight factorization* for training continuous neural representations across a range of tasks in computer vision, graphics, and scientific computing. More precisely, we compare the performance of plain MLPs, MLPs with adaptive activations (AA) (Jagtap et al., 2020), weight normalization (WN) (Salimans & Kingma, 2016), and the proposed random weight factorization (RWF). The comparison is performed over a collection of MLP architectures, including conventional MLPs, SIREN (Sitzmann et al., 2020), modified MLPs (Wang et al., 2021b), as well as MLPs with positional encodings (Mildenhall et al., 2020) and Gaussian Fourier features (Tancik et al., 2020). In addition, for the Image Regression and Computed Tomography benchmarks, we perform a comprehensive hyper-parameter sweep over different random seeds, learning rates, activation functions, scale hyper-parameters, as well as MLPs with different depths and widths. The hyper-parameters of our experiments are presented in Appendix E. Table 1 summarizes the results obtained for each benchmark, corresponding to the optimal input mapping and hyper-parameter setup. Overall, the proposed random weight factorization yields the best performance, significantly outperforming the other competing approaches.

It is also worth noting that the proposed moderately increases the model sizes by $\mathcal{O}(LD)$, where $L$ denotes the number of layers and $D$ the width of the network. Besides, the computational overhead

| Task | Metric | Case | Plain | AA | WN | RWF (ours) |
|------|--------|------|-------|-----|-----|-----------|
| Image Regression | PSNR (↑) | Natural | 27.35 | 27.37 | 27.36 | **28.08** |
| | | Text | 32.09 | 32.29 | 32.25 | **33.13** |
| Shape Representation | IoU (↑) | Dragon | 0.980 | 0.981 | 0.981 | **0.984** |
| | | Armadillo | 0.978 | 0.976 | 0.974 | **0.982** |
| Computed Tomography | PSNR (↑) | Shepp | 30.09 | 30.37 | 30.59 | **33.73** |
| | | ATLAS | 21.85 | 21.93 | 22.16 | **23.71** |
| Inverse Rendering | PSNR (↑) | Lego | 26.11 | 26.11 | 26.07 | **26.21** |
| Solving PDEs | Rel. $L^2$ (↓) | Advection | 25.94% | 36.27% | 67.40% | **3.10%** |
| | | Navier-Stokes | 11.93% | 10.95% | 14.62% | **6.79%** |
| Learning Operators | Rel. $L^2$ (↓) | DR | 1.09% | 0.95% | 0.97% | **0.50%** |
| | | Darcy | 2.03% | 2.06% | 2.00% | **1.67%** |
| | | Burgers | 5.11% | 4.71% | 4.37% | **2.46%** |

Table 1: We compare four different parameterizations over various benchmarks, and demonstrate that *Random weight factorization* consistently outperforms other parameterizations across all tasks. All comparisons are conducted under exactly the same hyper-parameter settings. (↑)/(↓) indicates that higher/lower values are better, respectively. (Plain: conventional MLP; AA: adatpive activation; WN: weight normalization; RWF: random weight factorization).

of our method is marginal, which is further clarified by Table 7 where we report the computational cost of each benchmark. Therefore, RWF can be therefore considered as a drop-in enhancement to any architecture that uses linear layers. All code and data will be made publicly available. A summary of each benchmark study is presented below, with more details provided in Appendix.

## 4.1 2D IMAGE REGRESSION

We train coordinate-based MLPs to learn a map from 2D input pixel coordinates to the corresponding RGB values of an image, using the benchmarks put forth in (Tancik et al., 2020). We conduct experiments using two data-sets: *Natural* and *Text*, each containing 16 images. The *Natural* data-set is constructed by taking center crops of randomly sampled images from the Div2K data-set (Agustsson & Timofte, 2017). The *Text* data-set is constructed by placing random text with random font sizes and colors on a white background. The training data is obtained by downsampling each test image by a factor of 2. We compare the resulting peak signal-to-noise ratio (PSNR) in the full resolution test images, obtained with different MLP architectures using different input mappings and weight parametrizations.

## 4.2 3D SHAPE REPRESENTATION

This task follows the original problem setup in (Tancik et al., 2020). The goal is to learn an implicit representation of a 3D shape using Occupancy networks (Mescheder et al., 2019), which take spatial coordinates as inputs and predict 0 for points outside a given shape and 1 for points inside the shape. We use two complex triangle meshes commonly used in computer graphics: *Dragon* and *Armadillo*. The training data is generated by randomly sampling points inside a bounding box and calculating their labels using the ground truth mesh. We evaluate the trained model performance using the Intersection over Union (IoU) metric on a set of points randomly sampled near the mesh surface to better highlight the different mappings' abilities to resolve fine details.

## 4.3 2D COMPUTED TOMOGRAPHY (CT)

This task follows the original problem setup in (Tancik et al., 2020). We train an MLP to learn a map from 2D pixel coordinates to a corresponding volume density at those locations. Two data-sets are considered: procedurally-generated Shepp-Logan phantoms (Shepp & Logan, 1974) and 2D brain images from the ATLAS data-set (Liew et al., 2018). Different from the previous tasks, we observe integral projections of a density field instead of direct measurements. The network is trained in an indirect supervised fashion by minimizing a loss between a sparse set of ground-truth integral projections and integral projections computed from the network's output. We use PSNR to

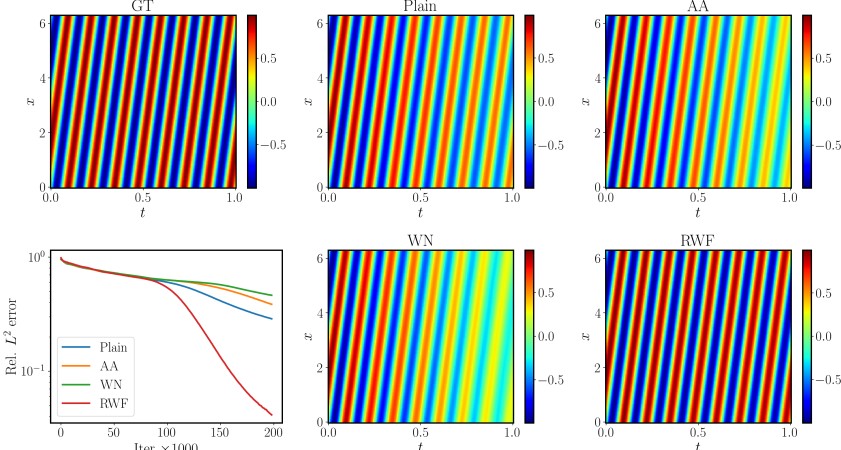

Figure 3: *Adection:* Predicted solutions of trained MLPs with different weight parameterizations, along with the evolution of the associated relative $L^2$ prediction errors during training.

quantify the performance of the trained MLPs with different input mappings and different weight parametrizations.

### 4.4 3D INVERSE RENDERING FOR VIEW SYNTHESIS

This task follows the original problem setup in (Tancik et al., 2020). We aim to learn an implicit representation of a 3D scene from 2D photographs using Neural Radiance Field (NeRF) (Mildenhall et al., 2020), which is a coordinate-based MLP that takes a 3D location as input and outputs a color and volume density. The network is trained by minimizing a rendering loss between the set of 2D image observations and the same rendered views from the predicted scene representation. In our experiments, we consider a down-sampled NeRF *Lego* data-set and use a simplified version of the method described in (Mildenhall et al., 2020), where we remove hierarchical sampling and view dependence. We compare the PSNR of the trained MLPs with different weight parametrizations over 10 random seeds.

### 4.5 SOLVING PARTIAL DIFFERENTIAL EQUATIONS (PDEs)

Our goal is to solve partial differential equations (PDEs) using physics-informed neural network (PINNs) (Raissi et al., 2019), which take the coordinates of a spatio-temporal domain as inputs and predict the corresponding target solution function. PINNs are trained in a self-supervised fashion by minimizing a composite loss function for fitting given initial and boundary conditions, as well as satisfying the underlying PDE constraints. We consider two benchmarks, an advection equation modeling the transport of a scalar field, and the Navier-Stokes equation modeling the motion of an incompressible fluid in a square cavity. We compare the resulting relative $L^2$ errors of each parameterizations over 10 random seeds. Detailed descriptions and implementations of each problem are provided below and in the Appendix K.

Figure 3 and Figure 4 present the ground truth against the predicted solutions obtained by training PINNs with different weight parameterizations. It can be observed that the predictions obtained by RWF are in excellent agreement with the ground truth, while the other three parameterizations result in poor or even non-physical approximations. The rapid decrease in the test error further validates the benefit of our method. We attribute these significant performance improvements to the fact that PINN models, due to their self-supervised nature, often suffer from poor initializations. Evidently, RWF can precisely mitigate this by being able to reach better local minima that are located further away from the model initialization neighborhood.

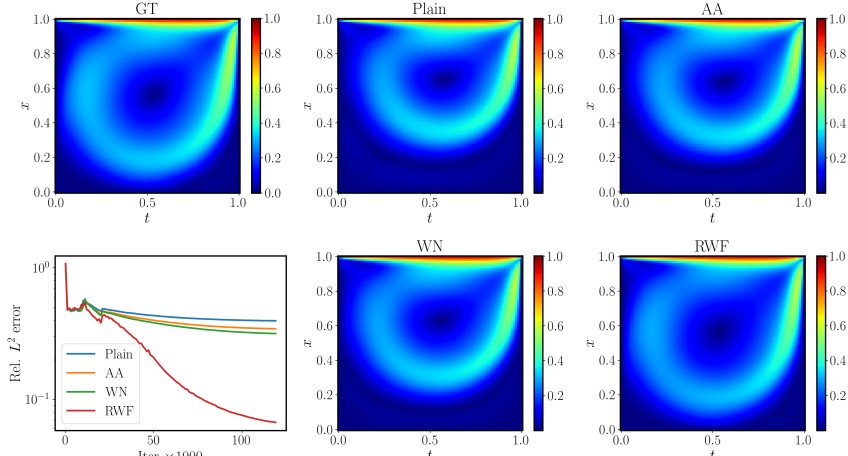

Figure 4: *Navier-Stokes:* Predicted solutions of trained MLPs with different weight parameterizations, along with the evolution of the associated relative $L^2$ errors during training.

**Advection equation:** The first example is 1D advection equation, a linear hyperbolic equation commonly used to model transport phenomena

$$\frac{\partial u}{\partial t} + c\frac{\partial u}{\partial x} = 0, \quad x \in (0, 2\pi), t \in [0, 1], \tag{4.1}$$

$$u(x, 0) = g(x), \quad x \in (0, 2\pi), \tag{4.2}$$

with periodic boundary conditions. This example has been studied in (Krishnapriyan et al., 2021; Daw et al., 2022), exposing some of the limitations that PINNs suffer from as the transport velocity $c$ is increased. In our experiments, we consider $c = 50$ and an initial condition $g(x) = \sin(x)$.

**Navier-Stokes equation:** The second example is a classical benchmark problem in computational fluid dynamics, describing the motion of an incompressible fluid in a two-dimensional lid-driven cavity. The system is governed by the Navier–Stokes equations written in a non-dimensional form

$$\boldsymbol{u} \cdot \nabla \boldsymbol{u} + \nabla p - \frac{1}{Re}\Delta \boldsymbol{u} = 0, \quad (x, y) \in (0, 1)^2, \tag{4.3}$$

$$\nabla \cdot \boldsymbol{u} = 0, \quad (x, y) \in (0, 1)^2, \tag{4.4}$$

where $\boldsymbol{u} = (u, v)$ denotes the velocity in $x$ and $y$ directions, respectively, and $p$ is the scalar pressure field. We assume $\boldsymbol{u} = (1, 0)$ on the top lid of the cavity, and a non-slip boundary condition on the other three walls. All experiments are performed with a Reynolds number of $Re = 1,000$.

## 4.6 LEARNING OPERATORS

In this task, we focus on learning the solution operators of parametric PDEs. To describe the problem setup in general, consider a parametric PDE of the following form

$$\mathcal{N}(\boldsymbol{a}, \boldsymbol{u}) = 0, \tag{4.5}$$

where $\mathcal{N} : \mathcal{A} \times \mathcal{U} \to \mathcal{V}$ is a linear or nonlinear differential operator between infinite-dimensional function spaces. Moreover, $\boldsymbol{a} \in \mathcal{A}$ denotes the PDE parameters, and $\boldsymbol{u} \in \mathcal{U}$ is the corresponding unknown solutions of the PDE system. The solution operator $G : \mathcal{A} \to \mathcal{U}$ is given by

$$G(\boldsymbol{a}) = \boldsymbol{u}(\boldsymbol{a}). \tag{4.6}$$

In our experiments, we consider three benchmarks: Diffusion-reaction, Darcy flow and the Burgers' equation. Detailed descriptions of each problem setup are shown below. As shown in Figure 5, we plot the training losses of each model with different weight parameterizations. One can see that *random weight factorization* yields the best loss convergence for every example, indicating the capability of the proposed method to accelerate the convergence of stochastic gradient descent and achieve better local minima.

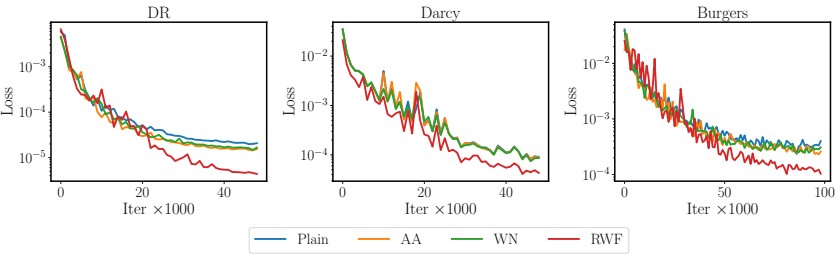

Figure 5: *Learning operators:* Loss convergence of training DeepONets with different weight parameterizations for diffusion-reaction equation, Darcy flow and the Burgers' equation.

**Diffusion-reaction:** Our first example involves a nonlinear diffusion-reaction PDE with a source term $a : (0, 1) \rightarrow \mathbb{R}$,

$$\frac{\partial u}{\partial t} = D\frac{\partial^2 u}{\partial x^2} + ku^2 + a(x), \quad (x, t) \in (0, 1) \times (0, 1], \tag{4.7}$$

with the zero initial and boundary conditions, where $D = 0.01$ is the diffusion coefficient and $k = 0.01$ is the reaction rate. We train a Deep Operator Network (DeepONet) Lu et al. (2021) to learn the solution operator for mapping source terms $a(x)$ to the corresponding PDE solutions $u(x)$. This network takes a PDE parameter and a spatial-temporal coordinate as inputs, and predicts the associated PDE solution evaluated at that location. The model is trained in a supervised manner by minimizing a loss between the predicted PDE solutions and the available solution measurements.

**Darcy flow:** The Darcy equation describes steady-state flow through a porous medium, taking the following form in two spatial dimensions

$$-\nabla \cdot (a \cdot \nabla u) = f, \quad (x, y) \in (0, 1)^2, \tag{4.8}$$

$$u = 0, \quad (x, y) \in \partial(0, 1)^2, \tag{4.9}$$

where $a : (0, 1) \rightarrow \mathbb{R}^+$ is the diffusion coefficient and $f : (0, 1) \rightarrow \mathbb{R}$ is a forcing term. We fix $f(x, y) = 1$ and aim to learn a continuous representation of the solution operator $G : a(x, y) \rightarrow u(x, y)$ with a DeepONet.

**Burgers' equation:** As the last example, we consider a fundamental nonlinear PDE, the one-dimensional viscous Burgers' equation. It takes the form

$$\frac{\partial u}{\partial t} + u\frac{\partial u}{\partial x} = \nu\frac{\partial^2 u}{\partial x^2}, \quad (x, t) \in (0, 1) \times (0, 1], \tag{4.10}$$

$$u(x, 0) = 0, \quad x \in (0, 1), \tag{4.11}$$

$$\tag{4.12}$$

with periodic boundary conditions and $\nu = 0.001$. Our goal is to learn the solution operator from the initial condition to the associated PDE solution with a physics-informed DeepONet (Wang et al., 2021e). Different from the first two examples, a physics-informed DeepONet is trained in a self-supervised manner, i.e. without any paired input-output observations, except for a set of given initial or boundary conditions (see Wang et al. (2021e) for more details).

## 5 CONCLUSIONS

In this work, we proposed *random weight factorization*, a simple and remarkably effective reparameterization of the weight matrices in neural networks. Theoretically, we show how this factorization alters the geometry of a loss landscape by assigning a self-adaptive learning rate to each neuron. Empirically, we show that our method can mitigate spectral bias in MLPs and enable networks to search for good local optima further away from their initialization. We validate *random weight factorization* using six different benchmarks ranging from image regression to learning operators, showcasing a consistent and robust improvements across various tasks in computer vision, graphics, and scientific computing. These findings provide new insights into the training of continuous neural representations and open several exciting avenues for future work, including the application of our method to deep learning models beyond coordinate-based MLPs, such as convolutional networks (LeCun et al., 1998), graph networks (Scarselli et al., 2008), and Transformers (Vaswani et al., 2017).

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

# A PROOFS

## A.1 PROOF OF THEOREM 1

*Proof.* Note that for any pair $(\boldsymbol{s}^{(l)}, V^{(l)})$ with $\mathrm{diag}(\boldsymbol{s}^{(l)}) \cdot \boldsymbol{V}^{(l)} = W^{(l)}$, we have

$$\|\boldsymbol{V}^{(l)}\| \longrightarrow 0, \text{ as } \boldsymbol{s}^{(l)} \longrightarrow \infty \tag{A.1}$$

for $l = 1, 2, \ldots, L+1$. Then for any $\epsilon > 0$, there exists $M > 0$ such that for any $(\boldsymbol{V}^{(l)}, \boldsymbol{s}^{(l)})_{l=1}^{L+1} \in U_{\boldsymbol{\theta}}$ ,and $\|\boldsymbol{s}^{(l)}\| > M$, we obtain $\|\boldsymbol{V}^{(l)}\| < \epsilon$, for $l = 1, 2, \ldots, L+1$. We define $U_*$ by

$$U_* = \{(\boldsymbol{0}, \boldsymbol{s}^{(l)})_{l=1}^{L+1} : \boldsymbol{s}^{(l)} \in \mathbb{R}^{d_l}, l = 1, 2, \ldots, L+1\} \tag{A.2}$$

Now we can choose $\boldsymbol{s}^{(l)}$ such that $\|\boldsymbol{s}^{(l)}\| > M$. Then

$$\mathrm{dist}(U_{\boldsymbol{\theta}}, U_*) \le \sqrt{\sum_{l=1}^{L+1} \|\boldsymbol{V}^{(l)}\|^2} \le \sqrt{L+1}\epsilon \tag{A.3}$$

Similarly, we can show that

$$\mathrm{dist}(U_{\boldsymbol{\theta}'}, U_*) \le \sqrt{\sum_{l=1}^{L+1} \|\boldsymbol{V}^{(l)}\|^2} \le \sqrt{L+1}\epsilon \tag{A.4}$$

Therefore,

$$\mathrm{dist}(U_{\boldsymbol{\theta}}, U_{\boldsymbol{\theta}'}) \le +\mathrm{dist}(U_{\boldsymbol{\theta}}, U_*) + \mathrm{dist}(U_*, U_{\boldsymbol{\theta}'}) \le 2\sqrt{L+1}\epsilon \tag{A.5}$$

Since $\epsilon$ is arbitrary, letting $\epsilon \to 0$ gives

$$\mathrm{dist}(U_{\boldsymbol{\theta}}, U_{\boldsymbol{\theta}'}) = 0. \tag{A.6}$$

$\square$

## A.2 PROOF OF THEOREM 2

*Proof.* Suppose that $f^{(k,l)}$ denotes $k$-th component of $\boldsymbol{f}^{(l)} \in \mathbb{R}^{d_l}$. Under the proposed weight factorization in equation 2.3, differentiating the loss function $\mathcal{L}$ with respect to $\boldsymbol{w}^{k,l}$ and $s^{(k,l)}$, respectively, yields

$$s_{n+1}^{(k,l)} = s_n^{(k,l)} - \eta \frac{\partial \mathcal{L}}{\partial s_n^{(k,l)}} = s_n^{(k,l)} - \eta \frac{\partial \mathcal{L}}{\partial f^{(k,l)}} \cdot \boldsymbol{v}_n^{(k,l)} \cdot \boldsymbol{g}^{(l-1)}, \tag{A.7}$$

$$\boldsymbol{v}_{n+1}^{(k,l)} = \boldsymbol{v}_n^{(k,l)} - \eta \frac{\partial \mathcal{L}}{\partial \boldsymbol{v}_n^{(k,l)}} = \boldsymbol{v}_n^{(k,l)} - \eta s_n^{(k,l)} \frac{\partial \mathcal{L}}{\partial f^{(k,l)}} \cdot \boldsymbol{g}^{(l-1)}. \tag{A.8}$$

Note that

$$\frac{\partial \mathcal{L}}{\partial \boldsymbol{w}_n^{(k,l)}} = \frac{\partial \mathcal{L}}{\partial f^{(k,l)}} \cdot \boldsymbol{g}^{(l-1)}, \tag{A.9}$$

and the update rule of $\boldsymbol{v}^{(k,l)}$ and $s^{(k,l)}$ can be re-written as

$$s_{n+1}^{(k,l)} = s_n^{(k,l)} - \eta \boldsymbol{v}_n^{(k,l)} \cdot \frac{\partial \mathcal{L}}{\partial \boldsymbol{w}_n^{(k,l)}}, \tag{A.10}$$

$$\boldsymbol{v}_{n+1}^{(k,l)} = \boldsymbol{v}_n^{(k,l)} - \eta s_n^{(k,l)} \frac{\partial \mathcal{L}}{\partial \boldsymbol{w}_n^{(k,l)}}. \tag{A.11}$$

Since $\boldsymbol{w}^{(k,l)} = s^{(k,l)} \cdot \boldsymbol{v}^{(k,l)}$, the update rule of $\boldsymbol{w}^{(k,l)}$ is given by

$$\boldsymbol{w}_{n+1}^{(k,l)} = \boldsymbol{w}_n^{(k,l)} - \eta \left( [s_n^{(k,l)}]^2 + \|\boldsymbol{v}_n^{(k,l)}\|_2^2 \right) \frac{\partial \mathcal{L}}{\partial \boldsymbol{w}_n^{(k,l)}} + \mathcal{O}(\eta^2) \tag{A.12}$$

$\square$

## B   ALGORITHM OF RANDOM WEIGHT FACTORIZATION

---

**Algorithm 1** Random weight factorization (RWF)

---

1. Initialize a neural network $f_{\boldsymbol{\theta}}$ with $\boldsymbol{\theta} = \{\boldsymbol{W}^{(l)}, \boldsymbol{b}^{(l)}\}_{l=1}^{L+1}$ (e.g. using the Glorot scheme (Glorot & Bengio, 2010)).
**for** $l = 1, 2, \ldots, L$ **do**
    (a) Initialize each scale factor as $\boldsymbol{s}^{(l)} \sim \mathcal{N}(\mu, \sigma I)$.
    (b) Construct the factorized weight matrices as $\boldsymbol{W}^{(l)} = \mathrm{diag}(\exp(\boldsymbol{s}^{(l)})) \cdot \boldsymbol{V}^{(l)}$.
**end for**
2. Train the network by gradient descent on the factorized parameters $\{\boldsymbol{s}^{(l)}, \boldsymbol{V}^{(l)}, \boldsymbol{b}^{(l)}\}_{l=1}^{L+1}$.
The recommended hyper-parameters are $\mu = 1.0, \sigma = 0.1$.

---

## C   A DROP-IN ENHANCEMENT FOR LINEAR LAYERS

```
class Dense(nn.Module):
    features: int

    @nn.compact
    def __call__(self, x):
        kernel = self.param('kernel',
                            glorot_normal(),
                            (x.shape[-1],
                             self.features))
        bias = self.param('bias',
                            nn.initializers.zeros,
                            (self.features,))
        y = np.dot(x, kernel) + bias
        return y
```

Listing 1: JAX Flax implementation (Heek et al., 2020) of a conventional linear layer.

```
def factorized_glorot_normal(mean=1.0, stddev=0.1):
    def init(key, shape):
        key1, key2 = random.split(key)
        w = glorot_normal()(key1, shape)
        s = mean + normal(stddev)(key2, (shape[-1],))
        s = np.exp(s)
        v = w / s
        return s, v
    return init

class FactorizedDense(nn.Module):
    features: int

    @nn.compact
    def __call__(self, x):
        s, v = self.param('kernel',
                            factorized_glorot_normal(),
                            (x.shape[-1], self.features))
        kernel = s * v
        bias = self.param('bias',
                            nn.initializers.zeros,
                            (self.features,))
        y = np.dot(x, kernel) + bias
        return y
```

Listing 2: JAX Flax implementation (Heek et al., 2020) of a linear layer with *random weight factorization*.

# D   1D REGRESSION ABLATION STUDY

We perform a systematic study on the effect of $\mu$ and $\sigma$ used for initializing the scale factor $s$ in *random weight factorization*. To this end, we train MLPs with *random weight factorization* initialized by different $\mu$ and $\sigma$. Each model (3 layers, 128 channels, ReLU activations) is trained via a full-batch gradient descent for $10^5$ iterations using the Adam optimizer (Kingma & Ba, 2014) with a starting learning rate of $10^{-3}$ followed by an exponential decay by a factor of 0.9 in every $5,000$ steps. The resulting relative $L^2$ errors over 10 random seeds are visualized in Figure 6. We observe that all the curves are roughly basin-shaped, indicating a "sweet spot" for the random weight factorization, i.e., $\mu \in [1, 2]$. Thus we will mainly use these as the default hyper-parameter of random weight factorization in the majority of the benchmarks presented here (see Table 5).

Furthermore, we perform an ablation study on the different distributions (Normal, LogNormal, Uniform, LogUniform) initializing the scale factors and report the average and standard deviation of mean square errors over 10 random seeds. For fair comparison, we manually tune each distribution such that the sampled values are roughly in the same range. We see that LogNormal is slightly better than the other distributions while overall there is no significant discrepancies in terms of the accuracy for using different distributions.

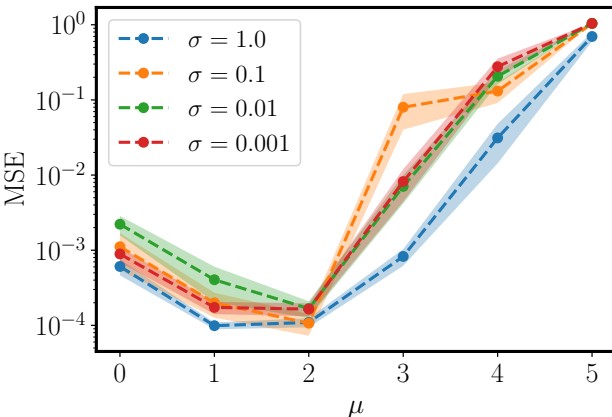

Figure 6: *1D Regression:* MSE of training MLPs with *random weight factorization* initialized by different $\mu$ and $\sigma$.

| Distribution | MSE |
|---|---|
| Normal | $3.95e - 04 \pm 2.22e - 04$ |
| LogNormal | $3.76e - 04 \pm 3.31e - 04$ |
| Uniform | $3.94e - 04 \pm 4.89e - 04$ |
| LogUniform | $5.40e - 04 \pm 4.78e - 04$ |

Table 2: *1D Regression:* MSE of training MLPs with *random weight factorization* initialized by different distributions.

# E   HYPER-PARAMETERS

The following tables summarizes the hyper-parameters of the different networks architectures employed in each benchmark (Table 3), their associated learning rate schedules (Table 4), and the corresponding random weight factorization settings (Table 5).

| Task | Case | Backbone | Depth | width | Activation |
|---|---|---|---|---|---|
| Image Regression | Natural | MLP | 4 | 256 | ReLU |
| | Text | | | | |
| Shape Representation | Dragon | MLP | 8 | 256 | ReLU |
| | Armadillo | | | | |
| Computed Tomography | Shepp | MLP | 5 | 256 | ReLU |
| | ATLAS | | | | |
| Inverse Rendering | Lego | MLP | 5 | 256 | ReLU |
| Solving PDEs | Advection | MLP | 5 | 256 | Tanh |
| | Navier-Stokes | MLP | 5 | 128 | |
| | | Modified MLP | | | |
| Learning Operators | DR | DeepONet | 5 | 64 | Tanh |
| | Darcy | | 4 | 128 | GELU |
| | Burgers | Modified DeepONet | 5 | 128 | Tanh |

Table 3: Network architectures for each benchmark in Table 1.

| Task | Case | Learning Rate Schedule | | | | Iterations |
|---|---|---|---|---|---|---|
| | | Step Size | Decay Steps | Decay Rate | Warmup Steps | |
| Image Regression | Natural | $10^{-3}$ | - | - | $2 \times 10^2$ | $2 \times 10^3$ |
| | Text | | | | | |
| Shape Representation | Dragon | $5 \times 10^{-4}$ | $5 \times 10^3$ | 0.1 | $10^3$ | $10^4$ |
| | Armadillo | | | | | |
| Computed Tomography | Shepp | $10^{-3}$ | - | - | $2 \times 10^2$ | $2 \times 10^3$ |
| | ATLAS | | | | | |
| Inverse Rendering | Lego | $10^{-3}$ | $10^3$ | 0.9 | $5 \times 10^3$ | $5 \times 10^4$ |
| Solving PDEs | Advection | $10^{-3}$ | $5 \times 10^3$ | 0.9 | - | $2 \times 10^5$ |
| | Navier-Stokes | | $2 \times 10^3$ | 0.9 | | $1.2 \times 10^5$ |
| Learning Operators | DR | $10^{-3}$ | $10^3$ | 0.9 | - | $5 \times 10^4$ |
| | Darcy | | | | | |
| | Burgers | | $2 \times 10^3$ | 0.9 | | $10^5$ |

Table 4: Learning Rate Schedules for each benchmark in Table 1.

| Task | Case | Initialization of RWF |
|------|------|----------------------|
| Image Regression | Natural | $s \sim \mathcal{N}(2, 0.01)$ |
| | Text | $s \sim \mathcal{N}(1, 0.1)$ |
| Shape Representation | Dragon | $s \sim \mathcal{N}(1, 0.1)$ |
| | Armadillo | |
| Computed Tomography | Shepp | $s \sim \mathcal{N}(1, 0.1)$ |
| | ATLAS | |
| Inverse Rendering | Lego | $s \sim \mathcal{N}(1, 0.1)$ |
| Solving PDEs | Advection | $s \sim \mathcal{N}(1, 0.1)$ |
| | Navier-Stokes | $s \sim \mathcal{N}(0.5, 0.1)$ |
| Learning Operators | DR | $s \sim \mathcal{N}(1, 0.1)$ |
| | Darcy | |
| | Burgers | |

Table 5: Distribution used for initializing the scale factor in *random weight factorization* for each benchmark.

| Hyper-parameters | Values |
|------------------|--------|
| Random Seed | 2, 3, 5, 7, 11 |
| Learning Rate | 1e-3, 1e-4 |
| Activation | ReLU, GELU, sin |
| Gaussian scales | 1, 5, 10 |
| Width | 64, 128, 256 |
| Depth | 3, 4, 5 |

Table 6: Range of hyper-parameters for hyper-parameter sweeps used in Image Regression and Computed Tomography benchmarks.

## F  COMPUTATIONAL COSTS

Table 7 presents the computational cost in terms of training iterations per second for the networks employed in each benchmark. All timings are reported on a single NVIDIA RTX A6000 GPU.

| Task | Case | Plain | AA | WN | RWF |
|------|------|-------|-----|-----|-----|
| Image Regression | Natural | 116.36 | 111.33 | 114.09 | 113.80 |
| | Text | 116.86 | 112.04 | 113.77 | 113.99 |
| Shape Representation | Dragon | 12.79 | 12.79 | 12.76 | 12.73 |
| | Armadillo | 13.14 | 13.06 | 12.81 | 12.94 |
| Computed Tomography | Shepp | 30.02 | 29.21 | 29.83 | 29.88 |
| | ATLAS | 29.47 | 28.74 | 29.36 | 29.32 |
| Inverse Rendering | Lego | 30.59 | 29.41 | 30.68 | 30.75 |
| Solving PDEs | Advection | 853.82 | 757.19 | 855.50 | 789.41 |
| | Navier-Stokes | 164.02 | 152.70 | 160.51 | 160.45 |
| Learning Operators | DR | 469.09 | 450.44 | 470.34 | 469.00 |
| | Darcy | 63.10 | 61.21 | 62.92 | 62.91 |
| | Burgers | 27.86 | 26.25 | 27.29 | 27.10 |

Table 7: Computational cost (training iterations per second) for each benchmark. We can see that the computational overhead of *random weight factorization* is marginal.

## G 2D IMAGE REGRESSION

As mentioned in Section 4.1, we use two image data-sets: *Natural* and *Natural*. All the test images have a $512 \times 512$ resolution while the training data has a $256 \times 256$ resolution. For each data-set, we train models with different weight parametrizations (Plain, Adaptive Activation, Weight Normalization, Random Weight Factorization) over different random seeds, learning rates, activation functions, scale hyper-parameters, as well as MLP architectures with different depths and widths. The range of hyper-parameters is summarized in Table 6. For random weight factorization, we initialize the scale factor $s \sim \mathcal{N}(2, 0.01)$ and $s \sim \mathcal{N}(1, 0.01)$ for the *Natural* and *Text* data-set, respectively. Each model is trained via a full-batch gradient descent for 2,000 iterations using the Adam optimizer (Kingma & Ba, 2014) with default settings and 200 warm-up steps. We compute the mean and standard deviations of the resulting PSNRs and visualize our results in Figure 7 and Figure 8, respectively. We reach the conclusion that the random weight factorization (RWF) consistently yields the best optimal performance than the alternatives.

Furthermore, under the best hyper-parameter setup, we compare the performance of MLPs with different parameterizations (see 2.3) and the following input embeddings:

**No mapping:** MLP with no input feature mapping.

**Positional encoding (Tancik et al., 2020):** $\gamma(\boldsymbol{x}) = \left[\ldots, \cos\left(2\pi\sigma^{j/m}\boldsymbol{x}\right), \sin\left(2\pi\sigma^{j/m}\boldsymbol{x}\right), \ldots\right]^{\mathrm{T}}$ for $j = 0, 1, \ldots, m - 1$. where the frequencies are log-linear spaced and the scale $\sigma > 0$ is a user-specified hyper-parameter.

**Gaussian (Tancik et al., 2020):** $\gamma(\boldsymbol{x}) = [\cos(2\pi\mathbf{B}\boldsymbol{x}), \sin(2\pi\mathbf{B}\boldsymbol{x})]^{\mathrm{T}}$, where $\mathbf{B} \in \mathbb{R}^{m \times d}$ is sampled from a Gaussian distribution $\mathcal{N}(0, \sigma^2)$. The scale $\sigma > 0$ is a user-specified hyper-parameter.

The resulting test PSNR is reported in Table 8 and Table 9. We can see that the weight factorization with Gaussian Fourier features achieves the best PSNR among all the cases. Some visualizations are shown in Figure 9 and Figure 10.

| *Natural* data-set | Plain | AA | WN | RWF (ours) |
|---|---|---|---|---|
| No mapping | $17.94 \pm 2.38$ | $18.29 \pm 2.44$ | $18.28 \pm 2.44$ | $\mathbf{19.11 \pm 2.50}$ |
| Positional Encoding | $27.04 \pm 3.89$ | $26.73 \pm 3.67$ | $26.99 \pm 3.83$ | $\mathbf{27.46 \pm 3.82}$ |
| Gaussian | $27.35 \pm 4.05$ | $27.36 \pm 3.96$ | $27.36 \pm 4.03$ | $\mathbf{28.08 \pm 4.34}$ |

Table 8: *2D Image Regression:* Mean and standard deviation of PSNR obtained by training MLPs with different input mappings and weight parameterizations for the *Natural* data-set.

| *Text* data-set | Plain | AA | WN | RWF (ours) |
|---|---|---|---|---|
| No mapping | $18.42 \pm 2.42$ | $18.43 \pm 2.34$ | $\mathbf{18.46 \pm 2.31}$ | $17.85 \pm 2.36$ |
| Positional encoding | $31.33 \pm 2.71$ | $\mathbf{31.73 \pm 2.29}$ | $31.43 \pm 2.57$ | $30.49 \pm 2.45$ |
| Gaussian | $32.09 \pm 1.80$ | $32.29 \pm 1.99$ | $32.25 \pm 1.74$ | $\mathbf{33.13 \pm 2.03}$ |

Table 9: *2D Image Regression:* Mean and standard deviation of PSNR obtained by training MLPs with different input mappings and weight parameterizations for the *Text* data-set.

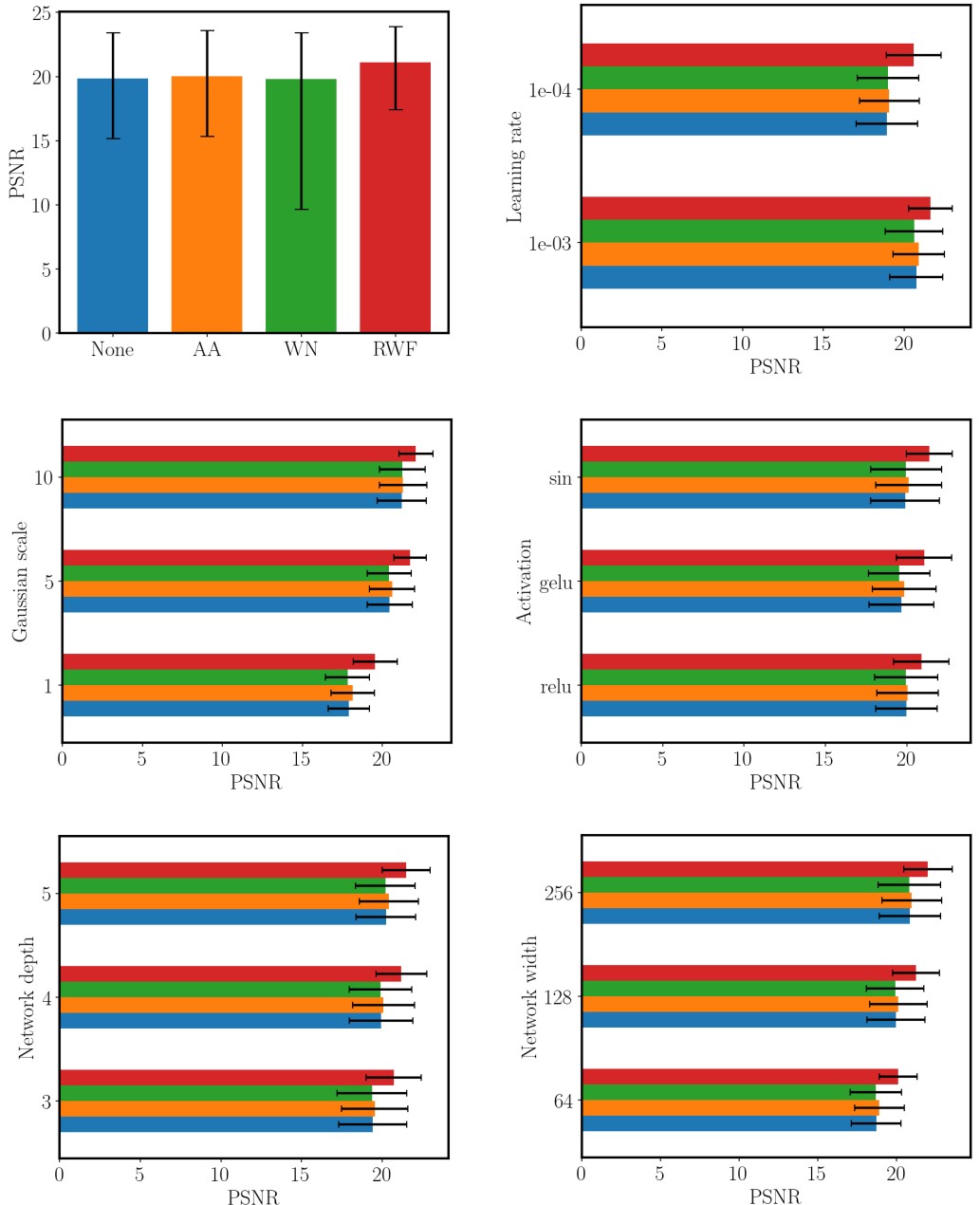

Figure 7: *2D Image Regression:* Mean and standard deviation of PSNR obtained by training MLPs with different hyper-parameter setup for the *Natural* data-set. In particular, top left panel plots a min/max bar, indicating that over the entire range of hyper-parameter settings tested, RWF yields the best optimal performance compared to the alternatives.

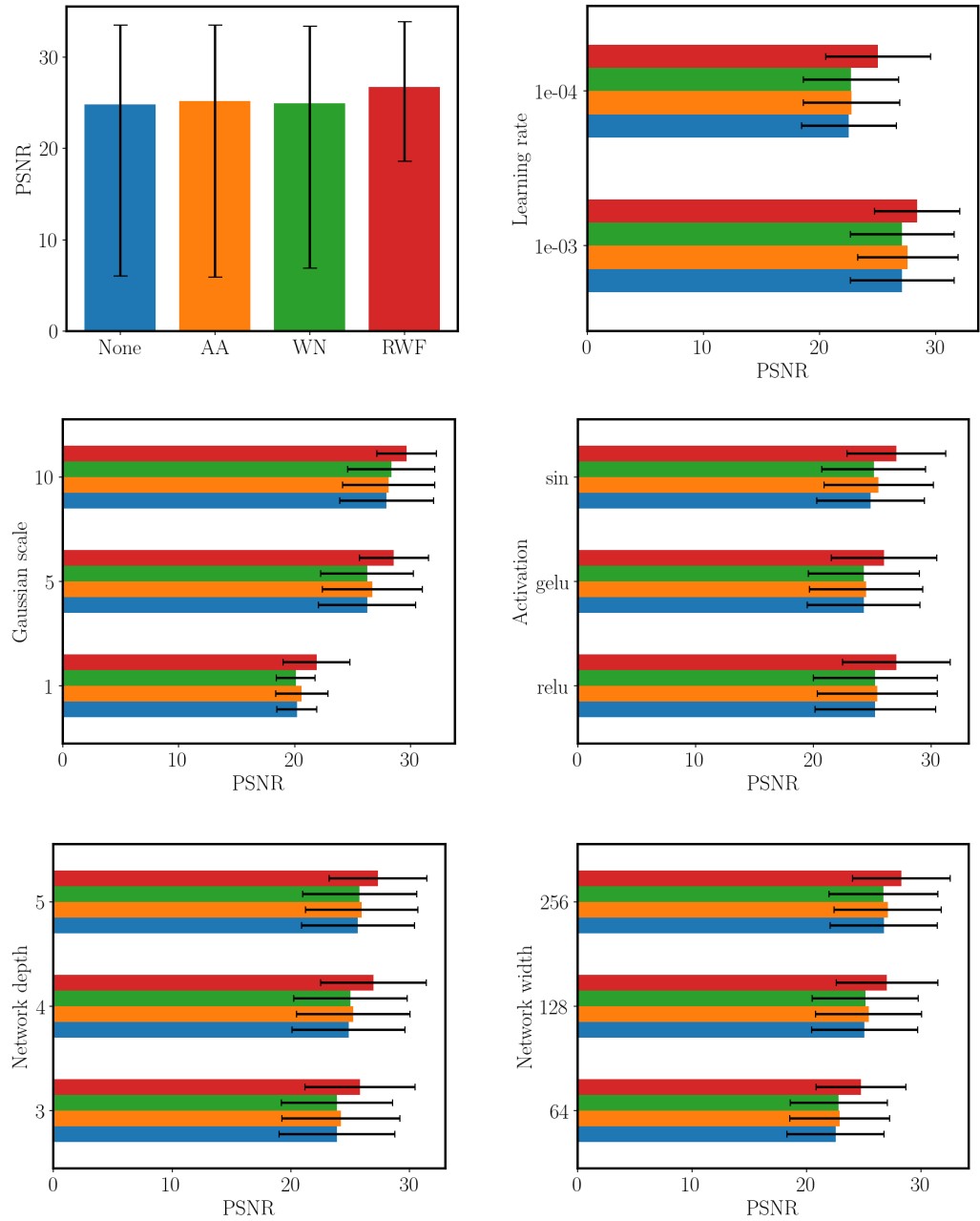

Figure 8: *2D Image Regression:* Mean and standard deviation of PSNR obtained by training MLPs with different hyper-parameter setup for the *Text* data-set. In particular, top left panel plots a min/-max bar, indicating that over the entire range of hyper-parameter settings tested, RWF yields the best optimal performance compared to the alternatives.

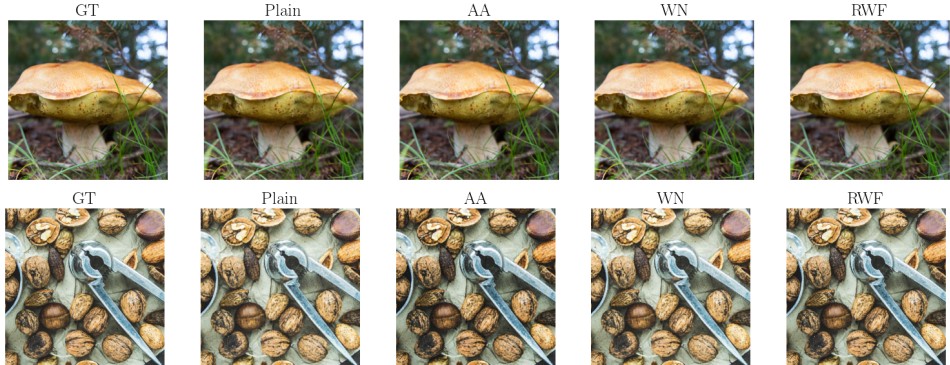

Figure 9: *2D Image Regression:* Predicted *Natural* images of trained MLPs with Gaussian Fourier features and with different weight parameterizations.

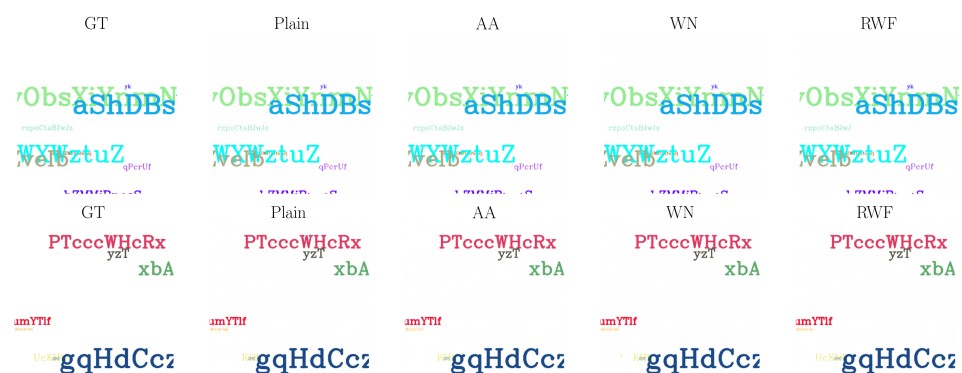

Figure 10: *2D Image Regression:* Predicted *Text* images of trained MLPs with Gaussian Fourier features and with different weight parameterizations.

**Comparison with SIREN (Sitzmann et al., 2020):** We find that SIREN also factorizes the weight matrix of every hidden layer as $W = \omega_0 \times \widehat{W}$ with some scale factor $\omega_0$. It is indeed a special case of our approach. To examine its performance, we vary the scale factor $w_0$ and train SIREN networks under the same hyper-parameter setting, and present the test error over the *Natural* and *Text* data-set in Figure 11. If we take $w_0 = 1$, the main difference between SIREN and our baseline is the activation function (sine vs. ReLU). However, SIREN just performs similarly to our baseline (Plain MLP with no input mapping). This suggests that the periodic activation function does not play an crucial role in improving the model performance. In the same figure, we can see that the SIREN performance is mostly affected by the value of $w_0$. Therefore, we may argue that the success of SIREN can be attributed to that simple weight factorization instead of the sine activations with the associated initialization scheme $W \sim \mathcal{U}(-\sqrt{6/d}, \sqrt{6/d})$. Nevertheless, the best PSNR that SIREN achieves is still significantly lower than the proposed random weight factorization with positional encodings or Gaussian Fourier features.

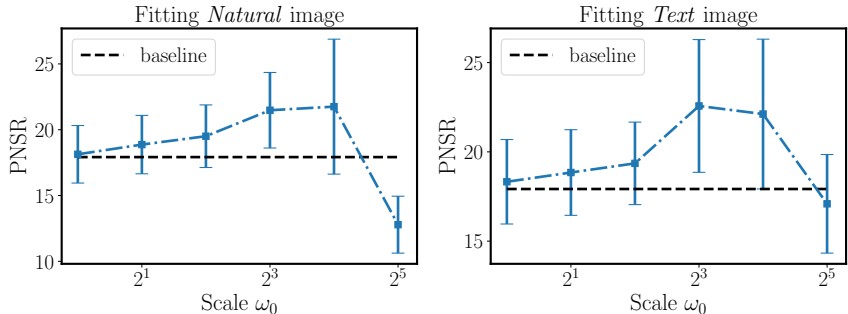

Figure 11: *2D Image Regression:* PSNR of training SIREN networks with different scale factor for the *Natural* and *Text* data-set, respectively. Error bars are plotted over different images in the data-set. The baseline (black dash) represents the result of training a plain MLP with no input mapping.

## H  2D COMPUTED TOMOGRAPHY

For this task, we use two data-sets: procedurally-generated Shepp-Logan phantoms (Shepp & Logan, 1974) and 2D brain images from the ATLAS data-set (Liew et al., 2018). Each data-set consist of 20 images of $512 \times 512$ resolution. To generate the training data, we compute 20 and 40 synthetic integral projections at evenly-spaced angles for every image of *Shepp* and *ATLAS* data-set, respectively.

Similar to the previous tasks, we first perform a hyper-parameter sweep over different random seeds, learning rates, activation functions, scale hyper-parameters, as well as MLP architectures with different depths and widths. Particularly, we initialize $s \sim \mathcal{N}(1, 0.1)$ for using random weight factorization. Each model is trained via a full-batch gradient descent for 2,000 iteration using the the Adam optimizer (Kingma & Ba, 2014) (Kingma & Ba, 2014) with default settings and 200 warm-up steps. Figure 12 and Figure 13 visualizes the the performance of MLPs with each weight parameterizations under different hyper-parameter settings.

Furthermore, under the best hyper-parameter setup, we compare the performance of MLPs with different input mappings. In experiments, Table 10 summarizes the test PSRN over *Shepp* and *ATLAS* data-set, respectively. For different input mappings, random weight factorization yields the best PSNR, consistently outperforming other parameterizations. Besides, we plot some model predictions corresponding to Gaussian input mapping in Figure 14 and 15.

| *Shepp* data-set | Plain | AA | WN | RWF (ours) |
|---|---|---|---|---|
| No mapping | $22.78 \pm 1.35$ | $23.44 \pm 1.27$ | $23.37 \pm 0.84$ | $\mathbf{24.39 \pm 1.56}$ |
| Positional encoding | $29.56 \pm 1.73$ | $29.71 \pm 1.80$ | $29.87 \pm 1.86$ | $\mathbf{32.43 \pm 1.51}$ |
| Gaussian | $30.08 \pm 1.63$ | $30.44 \pm 1.69$ | $30.55 \pm 1.72$ | $\mathbf{33.70 \pm 1.29}$ |
| *ATLAS* data-set | Plain | AA | WN | RWF (ours) |
| No mapping | $15.87 \pm 0.66$ | $16.07 \pm 0.68$ | $16.17 \pm 0.63$ | $\mathbf{16.49 \pm 0.61}$ |
| Positional encoding | $21.44 \pm 0.94$ | $21.54 \pm 0.92$ | $21.70 \pm 0.63$ | $\mathbf{23.34 \pm 0.92}$ |
| Gaussian | $22.02 \pm 0.93$ | $22.02 \pm 1.05$ | $22.10 \pm 1.04$ | $\mathbf{23.61 \pm 0.83}$ |

Table 10: *2D Computed Tomography:* Mean and standard deviation of PSNR obtained by training MLPs with different input mappings and weight parameterizations for the *Shepp* and *ATLAS* data-set, respectively.

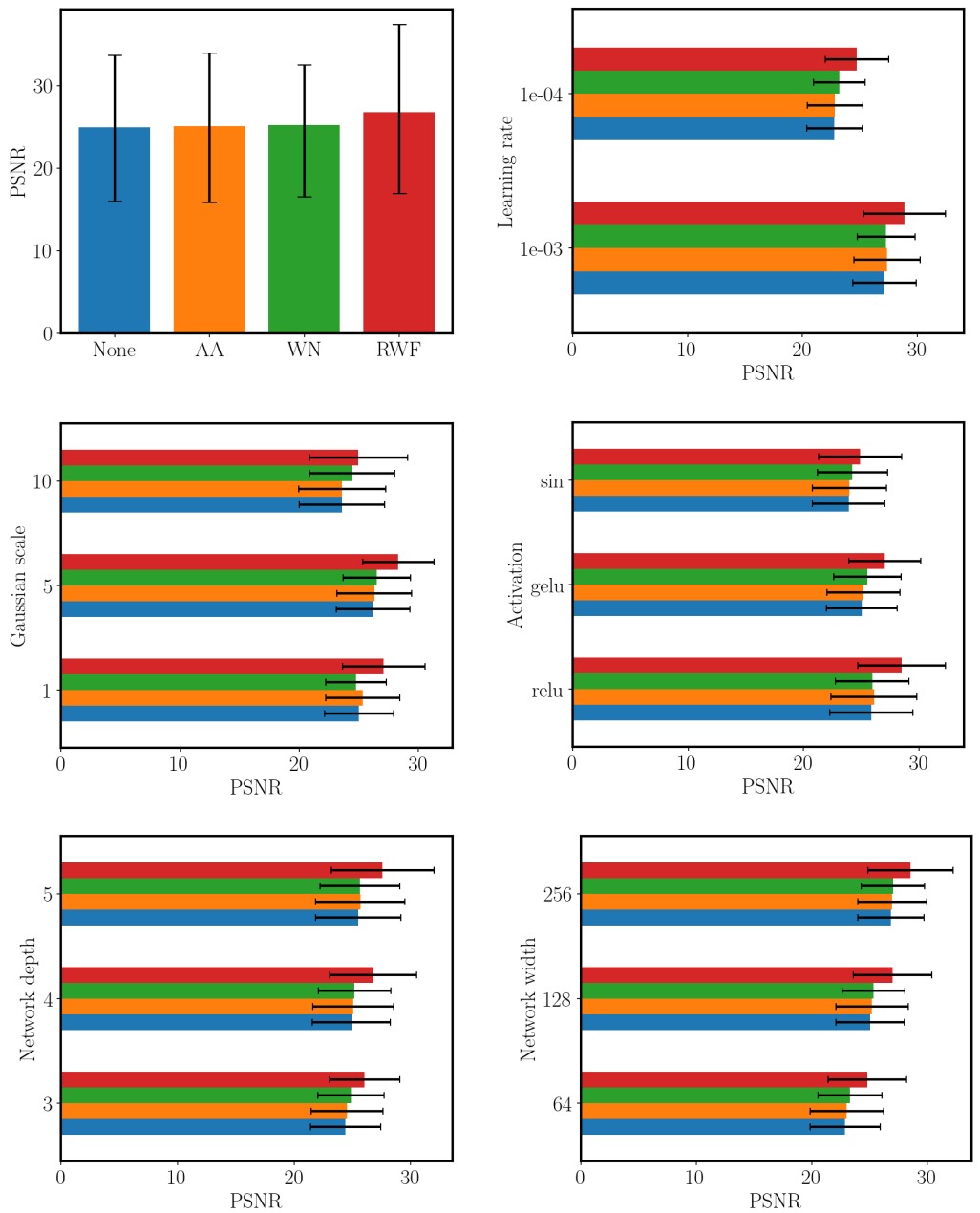

Figure 12: *2D Computed Tomography:* Mean and standard deviation of PSNR obtained by training MLPs with different hyper-parameter setup for the *Shepp* data-set. In particular, top left panel plots a min/max bar, indicating that over the entire range of hyper-parameter settings tested, RWF yields the best optimal performance compared to the alternatives.

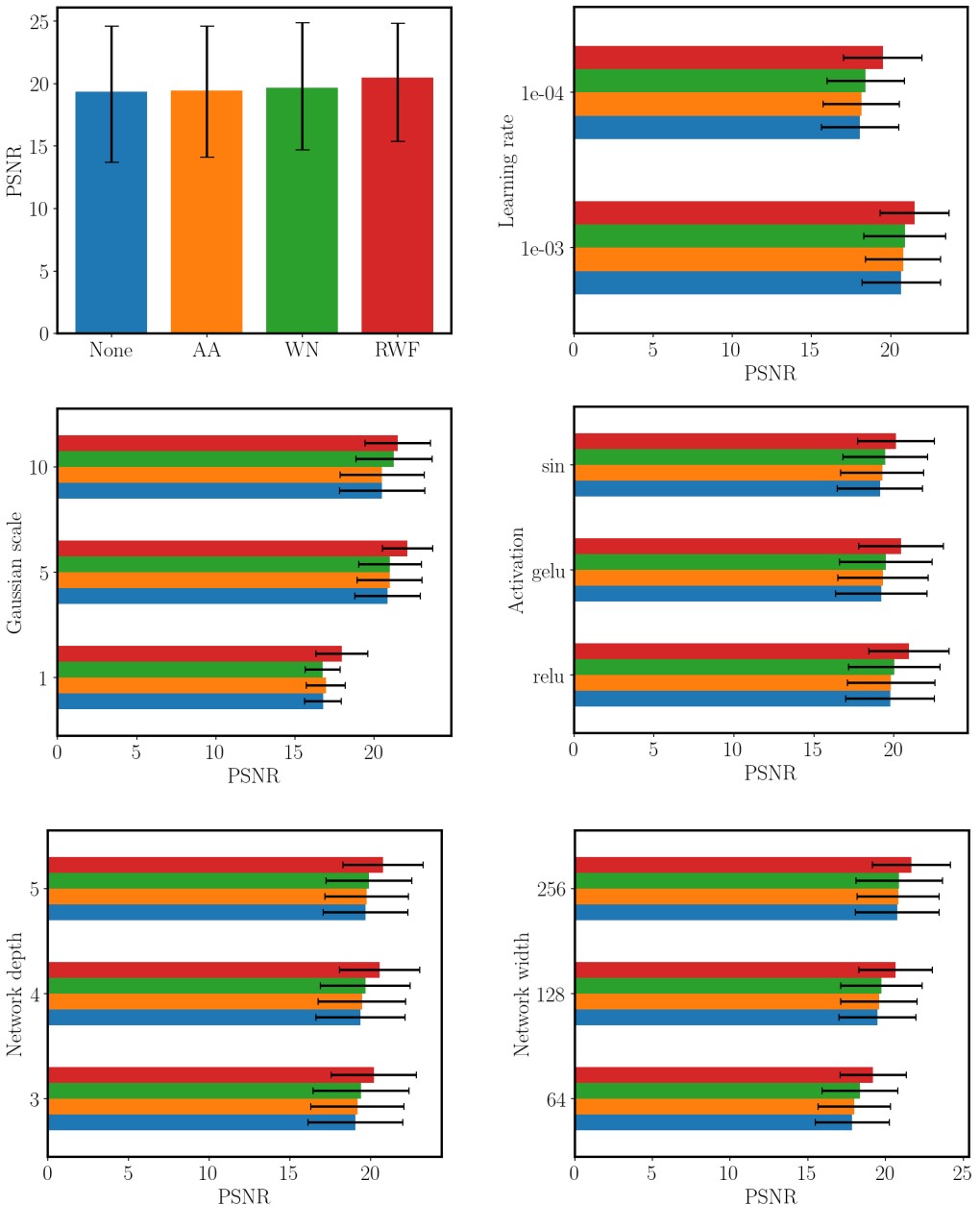

Figure 13: *2D Computed Tomography:* Mean and standard deviation of PSNR obtained by training MLPs with different hyper-parameter setup for the *Atlas* data-set. In particular, top left panel plots a min/max bar, indicating that over the entire range of hyper-parameter settings tested, RWF yields the best optimal performance compared to the alternatives.

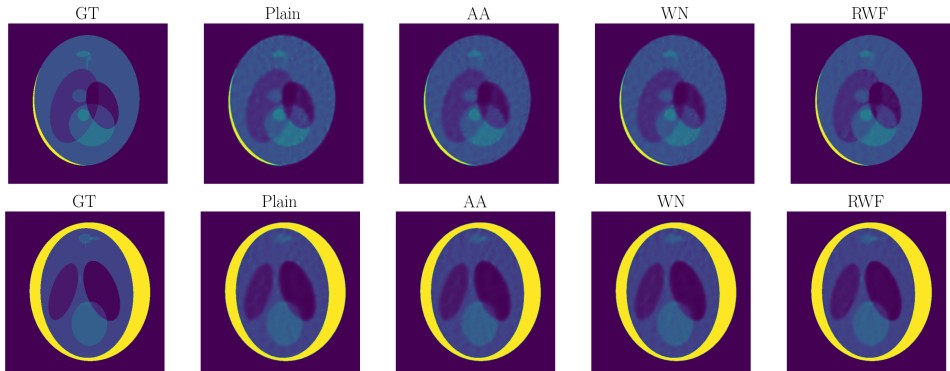

Figure 14: *2D Computed Tomography:* Predictions of trained MLPs with Gaussian Fourier features and with different weight parameterizations for the *Shepp* data-set.

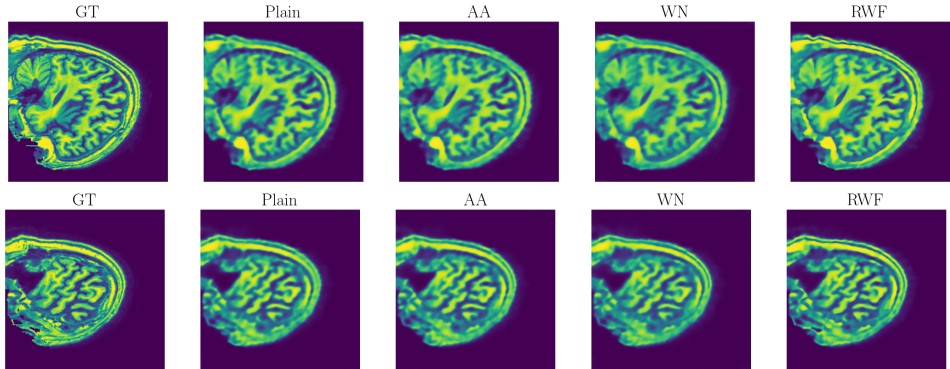

Figure 15: *2D Computed Tomography:* Predictions of trained MLPs with Gaussian Fourier features and with different weight parameterizations for the *ATLAS* data-set.

**Comparison with SIREN (Sitzmann et al., 2020):**  We also test the performance of SIREN for this example. Specifically, we train SIREN network with different scale factor $\omega_0$ under the same hyper-parameter settings. As shown in Figure 16, SIREN using the optimal scale factor is just slightly better than our baseline (plain MLP with no input mapping), but still worse than using positional encodings or random Fourier features.

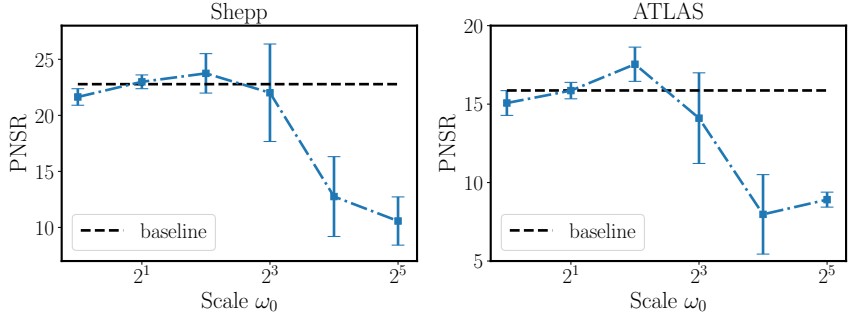

Figure 16: *2D Computed Tomography:* PSNR of training SIREN networks with different scale factor for the *Shepp* and *ATLAS* data-set, respectively. Error bars are plotted over different images in the data-set. The baseline (black dash) represents the result of training a plain MLP with no input mapping.

# I  3D SHAPE REPRESENTATION

For this example, we consider two complex triangle meshes *Dragon* and *Armadillo*, both of which contain hundreds of thousands of vertices. In our experiments, every mesh is rescaled to fit inside the unit cube $[0, 1]^3$ such that the centroid of the mesh is $(0.5, 0.5, 0.5)$.

We represent each shape by MLPs with different input mappings and weight parameterizations. For models with input mappings, we use the same hyper-parameters as in (Tancik et al., 2020). For models using *random weight factorization*, we initialize the scale factor using the recommended settings $s \sim \mathcal{N}(1, 0.1)$. All networks are trained by minimizing a cross-entropy loss to match the corresponding classification labels (0 for points outside the mesh, 1 for points inside).

We train each model (8 layers, 128 channels, ReLU activations) via a mini-batch gradient descent for $10^4$ iterations using the Adam optimizer (Kingma & Ba, 2014) with a start learning rate $5 \times 10^{-4}$ and an exponential decay by a factor of $0.1$ for every $5,000$ steps. The batch size we use is 8192. To emphasize the learning of fine surface details, we calculate the test error on a set close to the mesh surface, which is generated by randomly choosing mesh vertices that have been perturbed by a random Gaussian vector with a standard deviation of 0.01.

The resulting IoU scores of each model is reported in Table 11. One can observe consistent improvements of RWF across different input mappings and data-sets, outperforming the other parameterizations. Moreover, the learned shape representations are depicted in Figure 17.

| *Dragon* data-set | Plain | AA | WN | RWF (ours) |
|---|---|---|---|---|
| No mapping | 0.894 | 0.894 | 0.891 | **0.924** |
| Positional encoding | 0.967 | 0.968 | 0.970 | **0.977** |
| Gaussian | 0.980 | 0.981 | 0.981 | **0.984** |
| *Armadillo* data-set | Plain | AA | WN | RWF (ours) |
| No mapping | 0.842 | 0.846 | 0.845 | **0.901** |
| Positional encoding | 0.965 | 0.967 | 0.967 | **0.972** |
| Gaussian | 0.978 | 0.976 | 0.975 | **0.982** |

Table 11: *3D Shape Representation:* IoU of training MLPs with different input mappings and weight parameterizations for the *Dragon* and *Armadillo* data-sets.

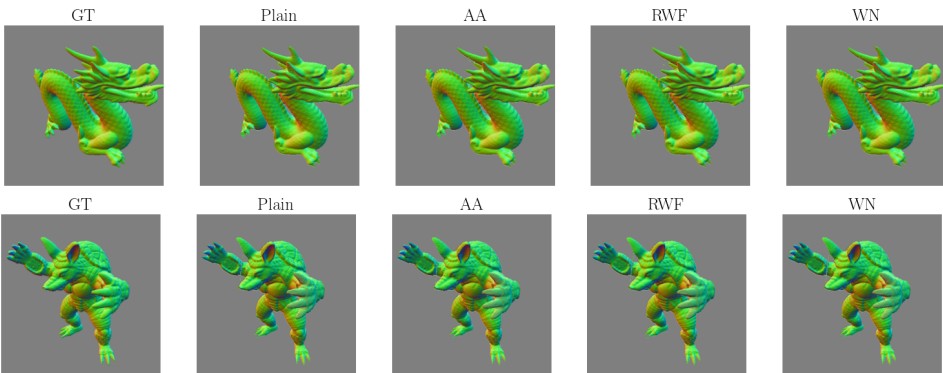

Figure 17: *3D Shape Representation:* Rendered shape representations obtained by training MLPs with Gaussian Fourier features and with different weight parameterizations.

# J  3D INVERSE RENDERING FOR VIEW SYNTHESIS

For this task, we use the NeRF *Lego* data-set of 120 images downsampled to $400 \times 400$ pixel resolution. The data-set is split into 100 training images, 7 validation images, and 13 test images. In

our experiments, we only use Gaussian Fourier features of a scale $\sigma = 10$, as it has been empirically validated to be the best input mapping in the previous tasks.

We train MLPs (5 layers, 256 channels, ReLU activations) with different parameterizations and 10 random seeds for $5 \times 10^4$ iterations over 10 random seeds. We use the Adam optimizer (Kingma & Ba, 2014) with a start learning rate of $10^{-3}$ and a warmup exponential decay by a factor of $0.9$ for every $1,000$ steps. The batch size is 2048. In Figure 18, we visualize the test PSNR of each model during training. In comparison with other three parameterizations, the MLP with RWF achieves the best PSRN. Some visualizations are provided in Figure 19

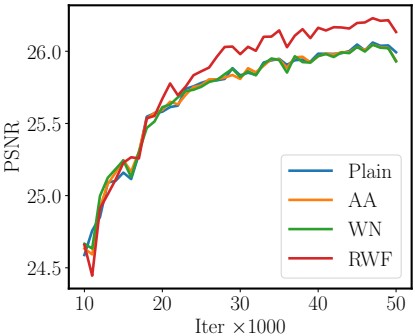

Figure 18: *3D Inverse Rendering:* Test PSNR of MLPs with different weight parameterizations during training.

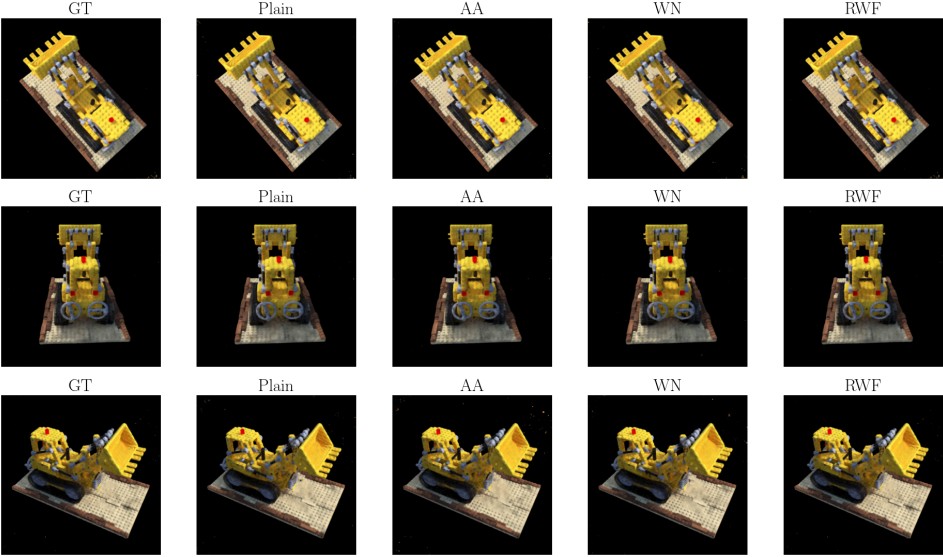

Figure 19: *3D Inverse Rendering:* Rendered views of trained MLPs with different weight parameterizations.

## K    SOLVING PDEs

In this section, we present the implementation details of PINNs for solving advection and Navier-Stokes equation, respectively.

### K.1 ADVECTION EQUATION

Recall

$$\frac{\partial u}{\partial t} + c\frac{\partial u}{\partial x} = 0, \quad x \in (0, 2\pi), t \in [0, 1] \tag{K.1}$$

$$u(x, 0) = g(x), \quad x \in (0, 2\pi) \tag{K.2}$$

with periodic boundary conditions and $c = 50$.

We represent the unknown solution $u$ by an MLP $u_{\boldsymbol{\theta}}$. In particular, we impose the exact periodic boundary conditions by constructing a special Fourier feature embedding of the form (Dong & Ni, 2021)

$$\gamma(x, t) = [\cos(x), \sin(x), t]^{\mathrm{T}}. \tag{K.3}$$

The network can be trained by minimizing the composite loss below

$$\mathcal{L}(\boldsymbol{\theta}) = \lambda_{ic}\mathcal{L}_{ic}(\boldsymbol{\theta}) + \lambda_r\mathcal{L}_r(\boldsymbol{\theta}) \tag{K.4}$$

where

$$\mathcal{L}_{ic}(\boldsymbol{\theta}) = \frac{1}{N_{ic}} \sum_{i=1}^{N_{ic}} \left| u_{\boldsymbol{\theta}}(0, x_{ic}^i) - g(x_{ic}^i) \right|^2, \tag{K.5}$$

$$\mathcal{L}_r(\boldsymbol{\theta}) = \frac{1}{N_r} \sum_{i=1}^{N_r} \left| \frac{\partial u_{\boldsymbol{\theta}}}{\partial t}(t_r^i, x_r^i) + c\frac{\partial u_{\boldsymbol{\theta}}}{\partial x}(t_r^i, x_r^i) \right|^2. \tag{K.6}$$

Here we set $N_{ic} = 128$ and $N_r = 1024$, and $\{x_{ic}\}_{i=1}^{N_{ic}}, \{(x_r, t_r)\}_{i=1}^{N_r}$ are randomly sampled from the computational domain, respectively, at each iteration of gradient descent. In addition, we take $\lambda_{ic} = 100, \lambda_r = 1$ for better enforcing the initial condition. It is worth pointing out that all the network derivatives are computed via automatic differentiation.

To enhance the model performance, we introduce the *curriculum* training (Krishnapriyan et al., 2021) and *causal* training (Wang et al., 2022) in the training process.

***Curriculum* training** starts with a simple PDE system and progressively solves the target PDE system. For this example, it is accomplished by minimizing the above PINN loss with a lower advection coefficient $c = 10$ first and then gradually increasing $c$ to the target value (i.e. $c = 50$) during training.

***Causal* training** aims to impose temporal causality during the training of a PINNs model by appropriately re-weighting the PDE residual loss at each iteration of gradient descent. Specifically, we split the temporal domain into $M$ chunks $[0, \Delta t], [\Delta t, 2\Delta t], \dots,$ and assign a weight to the corresponding temporal residuals losses $\mathcal{L}_r^i(\boldsymbol{\theta})$ as

$$\mathcal{L}_r(\boldsymbol{\theta}) = \sum_{i=1}^{M} w_i \mathcal{L}_i(\boldsymbol{\theta}), \tag{K.7}$$

with $w_1 = 1$, and

$$w_i = \exp(-\epsilon \sum_{k=1}^{i-1} \mathcal{L}_r^i(\boldsymbol{\theta})), \quad \text{for } i = 2, \dots, M. \tag{K.8}$$

Here $\epsilon$ is a so-called *causal* parameter, which is a user-specified hyper-parameter that determines the slope of the causal weights. We take $M = 16$ and $\epsilon = 0.1$ in this example.

We initialize MLPs (5 layers, 256 channels, tanh activations) with different weight parameterizations and 10 random seeds. We train each model with different strategies for $2 \times 10^5$ iterations using the Adam optimizer (Kingma & Ba, 2014) with a start learning rate of $10^{-3}$ and an exponential decay by a factor of 0.9 for every $5,000$ steps. The resulting relative $L^2$ errors are presented in Table 12. In contrast to the failure of the other three parameterizations, RWF is the only one that enables PINN models to solve the advection equation with a reasonable and stable predictive accuracy. Further improvements can be obtained by combining RWF with *curriculum* or *causal* training strategies.

These conclusions are further clarified by the visualizations in Figures 3, 20 and 21, where the predicted solutions corresponding to RWF are in excellent agreement with the ground truth.

| Advection | Plain | AA | WN | RWF (ours) |
|---|---|---|---|---|
| Regular | $25.94\% \pm 8.18\%$ | $36.27\% \pm 16.12\%$ | $67.40\% \pm 1.72\%$ | $\mathbf{3.10\% \pm 0.63\%}$ |
| Curriculum | $47.13\% \pm 20.40\%$ | $23.79\% \pm 25.86\%$ | $24.10\% \pm 18.85\%$ | $\mathbf{2.75\% \pm 1.74\%}$ |
| Causal | $5.92\% \pm 2.85\%$ | $10.22\% \pm 13.50\%$ | $37.38\% \pm 15.94\%$ | $\mathbf{2.08\% \pm 0.29\%}$ |

Table 12: Caption

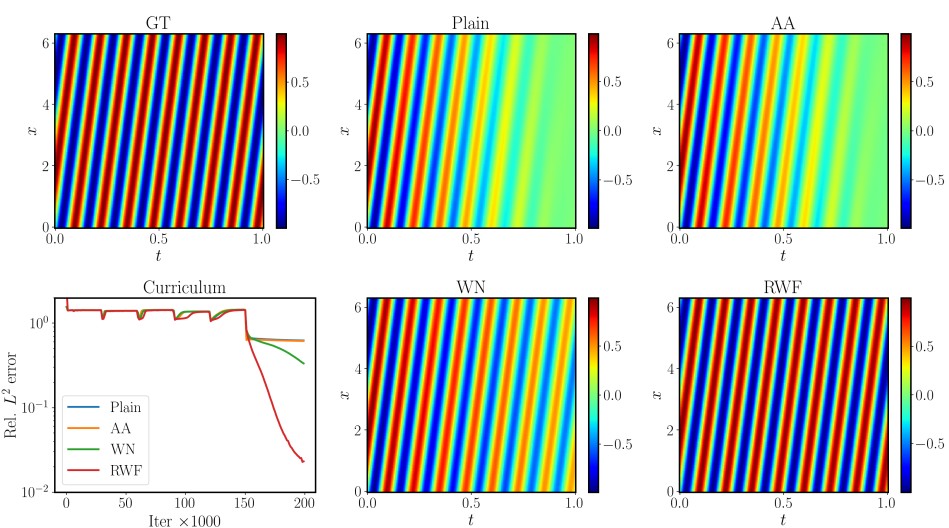

Figure 20: *advection:* Predicted solutions of training PINNs with different weight parameterizations using curriculum training, as well as the evolution of the associated relative $L^2$ errors during training.

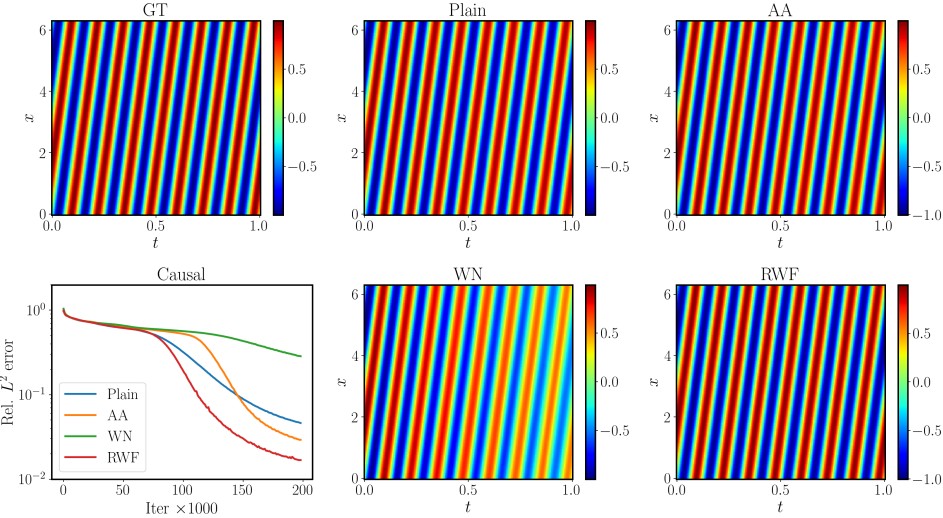

Figure 21: *advection:* Predicted solutions of training PINNs with different weight parameterizations using causal training, as well as the evolution of the associated relative $L^2$ errors during training.

## K.2 NAVIER-STOKES EQUATION

The underlying PDE system for this benchmark takes the form

$$\boldsymbol{u} \cdot \nabla \boldsymbol{u} + \nabla p - \frac{1}{Re} \Delta \boldsymbol{u} = 0, \quad (x, y) \in (0, 1)^2, \tag{K.9}$$

$$\nabla \cdot \boldsymbol{u} = 0, \quad (x, y) \in (0, 1)^2, \tag{K.10}$$

$$\boldsymbol{u} = (1, 0), \quad (x, y) \text{ on } \Gamma_1, \tag{K.11}$$

$$\boldsymbol{u} = (0, 0), \quad (x, y) \text{ on } \Gamma_2. \tag{K.12}$$

Here $\Gamma_1$ is the top boundary of a square cavity, while $\Gamma_2$ denotes the other three sides of the cavity. We represent the unknown solution $u, v, p$ using a neural network $\boldsymbol{u_\theta}$:

$$[x, y] \xrightarrow{\boldsymbol{u_\theta}} [u_{\boldsymbol{\theta}}, v_{\boldsymbol{\theta}}, p_{\boldsymbol{\theta}}]. \tag{K.13}$$

Then, the PDE residuals are defined by

$$\mathcal{R}_{\boldsymbol{\theta}}^u = u_{\boldsymbol{\theta}} \frac{\partial u_{\boldsymbol{\theta}}}{\partial x} + v_{\boldsymbol{\theta}} \frac{\partial u_{\boldsymbol{\theta}}}{\partial y} + \frac{\partial p_{\boldsymbol{\theta}}}{\partial x} - \frac{1}{\text{Re}} \left( \frac{\partial^2 u_{\boldsymbol{\theta}}}{\partial x^2} + \frac{\partial^2 u_{\boldsymbol{\theta}}}{\partial y^2} \right), \tag{K.14}$$

$$\mathcal{R}_{\boldsymbol{\theta}}^v = u_{\boldsymbol{\theta}} \frac{\partial v_{\boldsymbol{\theta}}}{\partial x} + u_{\boldsymbol{\theta}} \frac{\partial v_{\boldsymbol{\theta}}}{\partial y} + \frac{\partial p_{\boldsymbol{\theta}}}{\partial y} - \frac{1}{\text{Re}} \left( \frac{\partial^2 u_{\boldsymbol{\theta}}}{\partial x^2} + \frac{\partial^2 u_{\boldsymbol{\theta}}}{\partial y^2} \right), \tag{K.15}$$

$$\mathcal{R}_{\boldsymbol{\theta}}^c = \frac{\partial u_{\boldsymbol{\theta}}}{\partial x} + \frac{\partial v_{\boldsymbol{\theta}}}{\partial y}. \tag{K.16}$$

Given these residuals, along with a set of appropriate boundary conditions, we can now formulate a loss function for training a physics-informed neural network as

$$\mathcal{L}(\theta) = \lambda_u \mathcal{L}_u(\theta) + \lambda_v \mathcal{L}_v(\theta) + \lambda_{r_u} \mathcal{L}_{r_u}(\theta) + \lambda_{r_v} \mathcal{L}_{r_v}(\theta) + \lambda_{r_c} \mathcal{L}_{r_c}(\theta), \tag{K.17}$$

with

$$\mathcal{L}_{u_b}(\theta) = \frac{1}{N_b} \sum_{i=1}^{N_b} \left[ u \left( x_b^i, y_b^i \right) - u_b^i \right]^2, \tag{K.18}$$

$$\mathcal{L}_{v_b}(\theta) = \frac{1}{N_b} \sum_{i=1}^{N_b} \left[ v \left( x_b^i, y_b^i \right) - v_b^i \right]^2, \tag{K.19}$$

$$\mathcal{L}_{r_u}(\theta) = \frac{1}{N_r} \sum_{i=1}^{N_r} \left[ R_{\boldsymbol{\theta}}^u \left( x_r^i, y_r^i \right) \right]^2, \tag{K.20}$$

$$\mathcal{L}_{r_v}(\theta) = \frac{1}{N_r} \sum_{i=1}^{N_r} \left[ R_{\boldsymbol{\theta}}^v \left( x_r^i, y_r^i \right) \right]^2, \tag{K.21}$$

$$\mathcal{L}_{r_c}(\theta) = \frac{1}{N_r} \sum_{i=1}^{N_r} \left[ R_{\boldsymbol{\theta}}^c \left( x_r^i, y_r^i \right) \right]^2, \tag{K.22}$$

where $\left\{ \left( x_b^i, y_b^i \right), u_b^i \right\}_{i=1}^{N_b}$ and $\left\{ \left( x_b^i, y_b^i \right), v_b^i \right\}_{i=1}^{N_b}$ denote the boundary data for the two velocity components at the domain boundaries $\Gamma_1$ and $\Gamma_2$, respectively, while $\left\{ \left( x_r^i, y_r^i \right) \right\}_{i=1}^{N_r}$ is a set of collocation points for enforcing the PDE constraints. All of them are sampled randomly at each iteration of gradient descent. In experiments, we set $N_b = 256, N_r = 1024$ and $\lambda_u = \lambda_v = 100, \lambda_{r_u} = \lambda_{r_v} = \lambda_{r_c} = 1$.

We employ an MLP (5 layers, 128 channels, tanh activations) to represent the latent variables of interest, and train the network with different parameterizations for $10^5$ iterations using the Adam optimizer (Kingma & Ba, 2014) with a start learning rate of $10^{-3}$ and an exponential decay by a factor of 0.9 for every $2,000$ training iterations. Moreover, we use a modified MLP architecure (see definition below) and curriculum training to enhance the model stability and performance. For the curriculum training, we minimize the loss with $Re = 100$ and $Re = 500$ for $2 \times 10^4$ iterations sequentially and change the Reynolds number to $Re = 1,000$ for the rest of the training. The

resulting relative $L^2$ errors over 10 random seeds are reported in Table 13. We can see that RWF performs the best among all parameterizations by a large margin. Some visualizations are shown in Figure 4 and Figure 22. We attribute this significant performance improvements to the fact that PINN models often suffer from poor initializations, and RWF precisely mitigates this by being able to reach better local minima that are located further away from the model initialization neighborhood.

| Navier-Stokes | Plain | AA | WN | RWF (ours) |
|---|---|---|---|---|
| MLP | $11.93\% \pm 4.59\%$ | $10.95\% \pm 2.76\%$ | $14.62\% \pm 4.04\%$ | $\mathbf{6.79\% \pm 1.25\%}$ |
| Modified MLP | $3.66\% \pm 0.51\%$ | $3.97\% \pm 0.42\%$ | $3.80\% \pm 3.97\%$ | $\mathbf{2.61\% \pm 1.92\%}$ |

Table 13: *Navier-Stokes:* Relative $L^2$ errors of training conventional MLPs and modified MLPs with different weight parameterizations, respectively.

**Modified MLP:** In (Wang et al., 2021b) Wang *et al.* proposed a novel architecture that was demonstrated to outperform conventional MLPs across a variety of PINNs benchmarks. Here, we will refer to this architecture as "modified MLP". The forward pass of a $L$-layer modified MLP is defined as follows

$$\boldsymbol{U} = \sigma(\boldsymbol{W}_1 \boldsymbol{x} + \boldsymbol{b}_1), \ \ \boldsymbol{V} = \sigma(\boldsymbol{W}_2 \boldsymbol{x} + \boldsymbol{b}_2), \tag{K.23}$$

$$\boldsymbol{H}^{(1)} = \sigma(\boldsymbol{W}^{(1)} \boldsymbol{x} + \boldsymbol{b}^{(1)}), \tag{K.24}$$

$$\boldsymbol{Z}^{(l)} = \sigma(\boldsymbol{W}^{(l+1)} \boldsymbol{H}^{(k)} + \boldsymbol{b}^{(l+1)}), \ \ l = 1, \ldots, L-1, \tag{K.25}$$

$$\boldsymbol{H}^{(l+1)} = (1 - \boldsymbol{Z}^{(l)}) \odot \boldsymbol{U} + \boldsymbol{Z}^{(l)} \odot \boldsymbol{V}, \ \ l = 1, \ldots, L-1, \tag{K.26}$$

$$\boldsymbol{u_\theta}(\boldsymbol{x}) = \boldsymbol{W}^{(L+1)} \boldsymbol{H}^{(L)} + \boldsymbol{b}^{(L+1)}, \tag{K.27}$$

where $\sigma$ denotes a nonlinear activation function, $\odot$ denotes a point-wise multiplication. All trainable parameters are given by

$$\boldsymbol{\theta} = \{\boldsymbol{W}_1, \boldsymbol{b}_1, \boldsymbol{W}_2, \boldsymbol{b}_1, (\boldsymbol{W}^{(l)}, \boldsymbol{b}^{(l)})_{l=1}^{L+1}\}. \tag{K.28}$$

This architecture is almost the same as a standard MLP network, with the addition of two encoders and a minor modification in the forward pass. Specifically, the inputs $\boldsymbol{x}$ are embedded into a feature space via two encoders $\boldsymbol{U}, \boldsymbol{V}$, respectively, and merged in each hidden layer of a standard MLP using a point-wise multiplication.

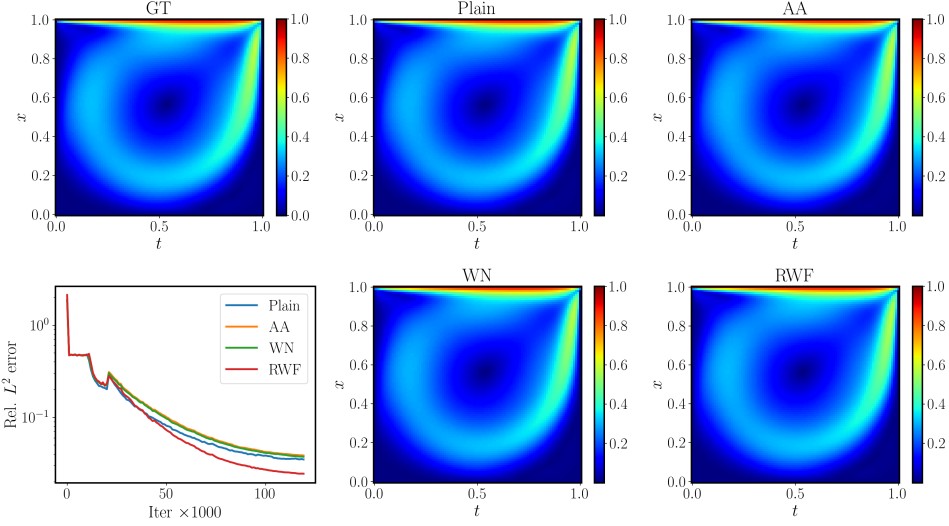

Figure 22: *Navier-Stokes:* Predicted solutions obtained by modified MLPs with different weight parameterization, as well as the evolution of the associated relative $L^2$ errors during training.

## L  LEARNING OPERATORS

**Overview of DeepONets:**  DeepONet is supposed to approximate an operator $G$ between functional spaces. DeepONet consists of two separate neural networks referred to as the "branch net" and "trunk net", respectively. The branch net takes a vector $\boldsymbol{a}$ as input and returns a features embedding $[b_1, b_2, \ldots, b_q]^T \in \mathbb{R}^q$ as output, where $\boldsymbol{a} = [a(\boldsymbol{x}_1), a(\boldsymbol{x}_2), \ldots, a(\boldsymbol{x}_m)]$ represents a function $a$ evaluated at a collection of fixed locations $\{\boldsymbol{x}_i\}_{i=1}^m$. The trunk net takes the continuous coordinates $\boldsymbol{y}$ as inputs, and outputs a features embedding $[t_1, t_2, \ldots, t_q]^T \in \mathbb{R}^q$. The DeepONet output is obtained by merging the outputs of the branch and trunk networks together via a dot product

$$G_{\boldsymbol{\theta}}(\boldsymbol{a})(\boldsymbol{y}) = \sum_{k=1}^q \underbrace{b_k\left(\boldsymbol{a}\left(\boldsymbol{x}_1\right), \boldsymbol{a}\left(\boldsymbol{x}_2\right), \ldots, \boldsymbol{a}\left(\boldsymbol{x}_m\right)\right)}_{\text{branch}} \underbrace{t_k(\boldsymbol{y})}_{\text{trunk}}, \tag{L.1}$$

where $\boldsymbol{\theta}$ denotes the collection of all trainable weight and bias parameters in the branch and trunk networks. These parameters can be optimized by minimizing the following mean square error loss

$$\mathcal{L}(\boldsymbol{\theta}) = \frac{1}{NP} \sum_{i=1}^N \sum_{j=1}^P \left| G_{\boldsymbol{\theta}}(\boldsymbol{a}^{(i)})(\boldsymbol{y}_j^{(i)}) - G(\boldsymbol{a}^{(i)})(\boldsymbol{y}_j^{(i)}) \right|^2 \tag{L.2}$$

$$= \frac{1}{NP} \sum_{i=1}^N \sum_{j=1}^P \left| \sum_{k=1}^q b_k(\boldsymbol{a}^{(i)}(\boldsymbol{x}_1), \ldots, \boldsymbol{a}^{(i)}(\boldsymbol{x}_m)) t_k(\boldsymbol{y}_j^{(i)}) - G(\boldsymbol{a}^{(i)})(\boldsymbol{y}_j^{(i)}) \right|^2, \tag{L.3}$$

where $\{\boldsymbol{a}^{(i)}\}_{i=1}^N$ denotes $N$ separate input functions sampled from a function space $\mathcal{U}$. For each $\boldsymbol{a}^{(i)}$, $\{\boldsymbol{y}_j^{(i)}\}_{j=1}^P$ are $P$ locations in the domain of $G(\boldsymbol{a}^{(i)})$, and $G(\boldsymbol{a}^{(i)})(\boldsymbol{y}_j^{(i)})$ is the corresponding output data evaluated at $\boldsymbol{y}_j^{(i)}$. Contrary to the fixed sensor locations of $\{x_i\}_{i=1}^m$, we remark that the locations of $\{\boldsymbol{y}^{(i)}\}_{j=1}^P$ may vary for different $i$, thus allowing us to construct a continuous representation of the output function $G(a)$.

**Remark:** All parameterizations (AA, WN, RWF) are applied to every dense layer of the DeepONet architecture (in the cases where such parametrizations are employed).

| Case | Plain | AA | WN | RWF |
|---|---|---|---|---|
| DR | $1.09\% \pm 0.53\%$ | $0.95\% \pm 0.46\%$ | $0.97\% \pm 0.46\%$ | $\mathbf{0.50\% \pm 0.26\%}$ |
| Darcy | $2.03\% \pm 1.54\%$ | $2.06\% \pm 1.70\%$ | $2.05\% \pm 2.00\%$ | $\mathbf{1.67\% \pm 1.70\%}$ |
| Burgers | $5.11\% \pm 3.79\%$ | $4.71\% \pm 3.39\%$ | $4.37\% \pm 2.63\%$ | $\mathbf{2.46\% \pm 2.05\%}$ |

Table 14: *Learning operators:* Relative $L^2$ errors of trained (physics-informed) DeepONets over the test data-set of different examples.

### L.1  DIFFUSION-REACTION

The underlying PDE for this benchmark takes the form

$$\frac{\partial u}{\partial t} = D\frac{\partial^2 u}{\partial x^2} + ku^2 + a(x), \quad (x,t) \in (0,1) \times (0,1], \tag{L.4}$$

subject to zero initial and boundary conditions.

**Data Generation:**  We sample $N = 5,000$ input functions $a(x)$ from a GRF with length scale $l = 0.2$ and solve the diffusion-reaction system using a second-order implicit finite difference method on a $100 \times 100$ equispaced grid. To generate the training data, we randomly take $P = 100$ measurements from each solution. The test data-set contains another 100 solutions evaluated at the same mesh.

We represent the solution operator by a DeepONet $G_{\boldsymbol{\theta}}$, where the branch and trunk networks are two separate MLPs (5 layers, 64 channels, tanh activations). The model is trained for $5 \times 10^4$ iterations using the Adam optimizer (Kingma & Ba, 2014) with a start learning rate of $10^{-3}$ and

an exponential decay by a factor of 0.9 for every 1000 steps. The mean and standard deviation of the relative $L^2$ errors over the test date-set are shown in Table 14. Figure 23 provides several representative predictions using RWF.

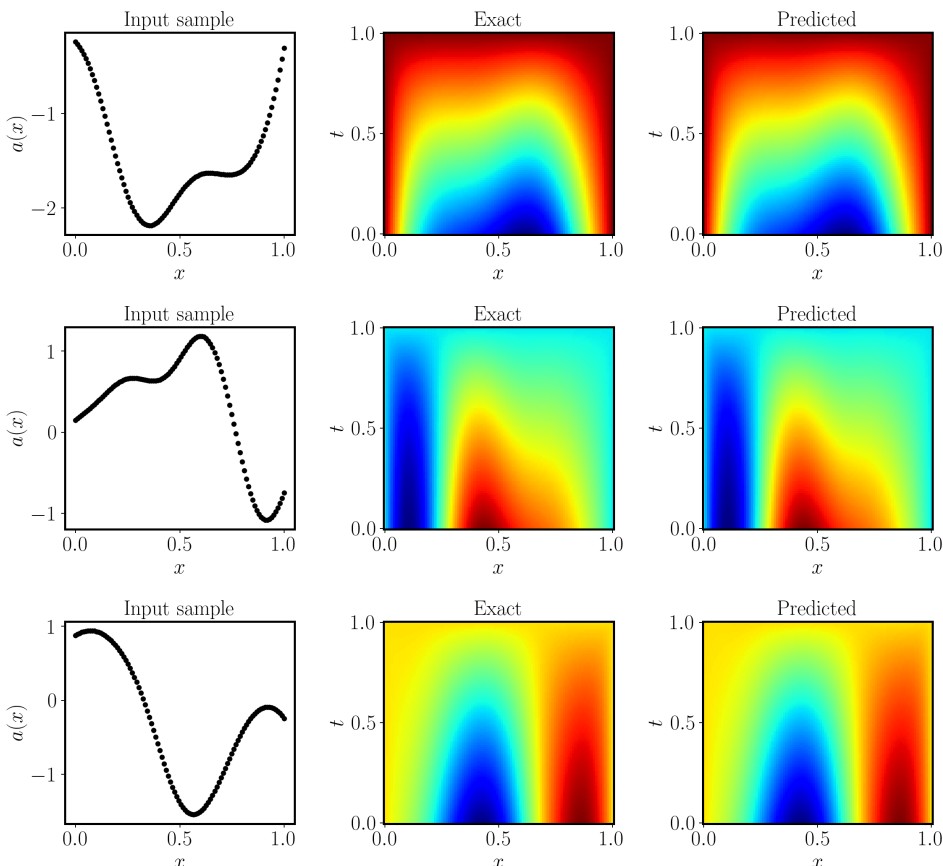

Figure 23: *Diffusion-reaction (DR) PDE:* Predicted solutions of a trained DeepONet with random weight factorization, corresponding to randomly chosen input samples in the test data-set.

## L.2 DARCY FLOW

The PDE system for this benchmark takes the form

$$-\nabla \cdot (a \cdot \nabla u) = 1, \quad (x, y) \in (0, 1)^2, \tag{L.5}$$

$$u = 0, \quad (x, y) \in \partial(0, 1)^2. \tag{L.6}$$

**Data Generation:** We sample the coefficient function $a$ from a Gaussian random field with a length scale $l = -0.5$ and solve the associated Darcy flow using finite element method. The training data contains $2,000$ solutions evaluated at a $64 \times 64$ uniform mesh while the test data contains $100$ solutions on the same mesh.

We represent the solution operator by a DeepONet $G_{\boldsymbol{\theta}}$, where the branch network is a convolutional neural network (CNN) for extracting latent feature representation of the input coefficients and the trunk network is a 4-layer MLP with GELU activations and 128 neurons per hidden layer. We train each model with different parameterizations for $5 \times 10^4$ iterations using the Adam optimizer (Kingma & Ba, 2014) with a start learning rate of $10^{-3}$ and an exponential decay by a factor of 0.9 for every 1000 steps. The results are summarized in Table 14 and some predicted solutions are plotted in Figure 24.

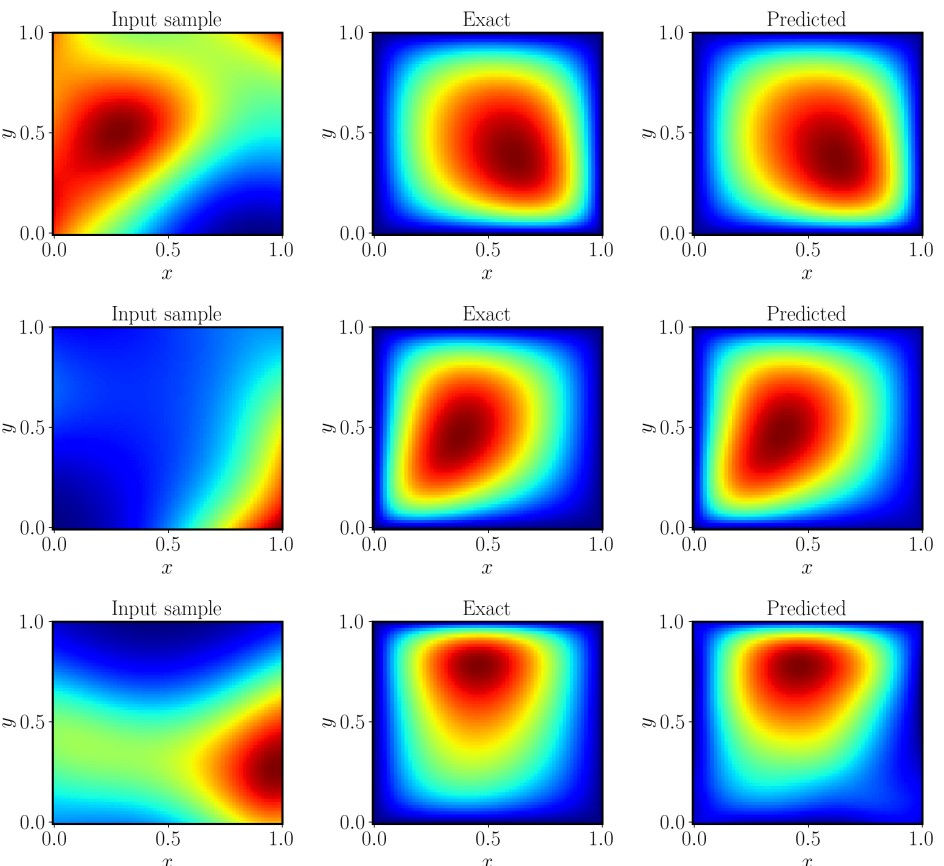

Figure 24: *Darcy PDE:* Predicted solutions of a trained DeepONet with random weight factorization, corresponding to randomly chosen input samples in the test data-set.

### L.3 BURGERS' EQUATION

Recall the one-dimensional Burgers' equation is given by

$$\frac{du}{dt} + u\frac{du}{dx} - \nu\frac{d^2u}{dx^2} = 0, \quad (x,t) \in (0,1) \times (0,1], \tag{L.7}$$

subject to initial and the periodic boundary conditions

$$u(x,0) = a(x), \quad x \in (0,1), \tag{L.8}$$

**Data Generation:** The training data only consists of $1,000$ input functions $a$ sampled from a Gaussian random field $\sim \mathcal{N}\left(0, 25^2(-\Delta + 5^2 I)^{-4}\right)$. To generate the test data-set, we sample another 100 input functions from the same Gaussian random field and solve the Burgers' equation using the Chebfun package Driscoll et al. (2014) with a spectral Fourier discretization and a fourth-order stiff time-stepping scheme (ETDRK4) Cox & Matthews (2002) with a time-step size of $10^{-4}$. Temporal snapshots of the solution are saved every $\Delta t = 0.01$ to give us 101 snapshots in total. Consequently, the test data-set contains 500 realizations evaluated at a $100 \times 101$ spatio-temporal grid.

Our objective here is to learn the solution operator mapping initial conditions $a(x)$ to the associated full spatio-temporal solution $u(x,t)$. Here proceed by representing the solution operator by a modified DeepONet architecture Wang et al. (2021d) outlined below. To impose the periodic the exact boundary condition, we apply a Fourier feature mapping to the input coordinates before passing them through the trunk network

$$[x,t] \rightarrow [\cos(2\pi x), \sin(2\pi x), t]. \tag{L.9}$$

Then the PDE residual is then defined by

$$R_{\boldsymbol{\theta}}[a] = \frac{\partial G_{\boldsymbol{\theta}}(\boldsymbol{a})}{\partial t} + G_{\boldsymbol{\theta}}(\boldsymbol{a})\frac{\partial G_{\boldsymbol{\theta}}(\boldsymbol{a})}{\partial x} - \nu\frac{\partial^2 G_{\boldsymbol{\theta}}(\boldsymbol{a})}{\partial x^2}, \tag{L.10}$$

Consequently, a physics-informed DeepONet can be trained by minimizing the following weighted loss function

$$\mathcal{L}(\boldsymbol{\theta}) = \lambda_{ic}\mathcal{L}_{ic}(\boldsymbol{\theta}) + \lambda_r\mathcal{L}_r(\boldsymbol{\theta}), \tag{L.11}$$

where

$$\mathcal{L}_{ic}(\boldsymbol{\theta}) = \frac{1}{NP}\sum_{i=1}^{N}\sum_{j=1}^{P}\left|G_{\boldsymbol{\theta}}(\boldsymbol{a}^{(i)})(x_{ic,j}^{(i)}, 0) - u^{(i)}(x_{ic,j}^{(i)})\right|^2, \tag{L.12}$$

$$\mathcal{L}_r(\boldsymbol{\theta}) = \frac{1}{NQ}\sum_{i=1}^{N}\sum_{j=1}^{Q}\left|R_{\boldsymbol{\theta}}^{(i)}(x_{r,j}^{(i)}, t_{r,j}^{(i)})\right|^2. \tag{L.13}$$

For this example, we take $N = 64, P = 100$ and $Q = 512$, which means that we randomly sample $N = 64$ input functions from the training data-set and $Q = 512$ collocation points inside the computational domain. In particular, we set $\lambda_{ic} = 100, \lambda_r = 1$ for better enforcing the initial condition across different input samples. The model with different parameterizations is trained for $10^5$ iterations using the the Adam optimizer (Kingma & Ba, 2014) with a start learning rate of $10^{-3}$ and an exponential decay by a factor of $0.9$ for every $2,000$ steps. We report the test errors in Table 14 and visualize some predicted solutions in Figure 25.

**Modified DeepONet:** Wang et al. (2021d) modify the forward pass of an L-layer DeepONet as follows

$$\boldsymbol{U} = \phi(\boldsymbol{W}_a\boldsymbol{a} + \boldsymbol{b}_a), \;\; \boldsymbol{V} = \phi(\boldsymbol{W}_y\boldsymbol{y} + \boldsymbol{b}_y), \tag{L.14}$$

$$\boldsymbol{H}_a^{(1)} = \phi(\boldsymbol{W}_a^{(1)}\boldsymbol{a} + \boldsymbol{b}_a^{(1)}), \;\; \boldsymbol{H}_y^{(1)} = \phi(\boldsymbol{W}_y^{(1)}\boldsymbol{y} + \boldsymbol{b}_y^{(1)}), \tag{L.15}$$

$$\boldsymbol{Z}_a^{(l)} = \phi(\boldsymbol{W}_a^{(l)}\boldsymbol{H}_a^{(l)} + \boldsymbol{b}_a^{(l)}), \;\; \boldsymbol{Z}_y^{(l)} = \phi(\boldsymbol{W}_y^{(l)}\boldsymbol{H}_y^{(l)} + \boldsymbol{b}_y^{(l)}), \quad l = 1, 2, \ldots, L-1, \tag{L.16}$$

$$\boldsymbol{H}_a^{(l+1)} = (1 - \boldsymbol{Z}_a^{(l)}) \odot \boldsymbol{U} + \boldsymbol{Z}_a^{(l)} \odot \boldsymbol{V}, \quad l = 1, \ldots, L-1, \tag{L.17}$$

$$\boldsymbol{H}_y^{(l+1)} = (1 - \boldsymbol{Z}_y^{(l)}) \odot \boldsymbol{U} + \boldsymbol{Z}_y^{(l)} \odot \boldsymbol{V}, \quad l = 1, \ldots, L-1, \tag{L.18}$$

$$\boldsymbol{H}_a^{(L)} = \phi(\boldsymbol{W}_a^{(L)}\boldsymbol{H}_a^{(L-1)} + \boldsymbol{b}_a^{(L)}), \;\; \boldsymbol{H}_y^{(L)} = \phi(\boldsymbol{W}_y^{(L)}\boldsymbol{H}_y^{(L-1)} + \boldsymbol{b}_y^{(L)}), \tag{L.19}$$

$$G_{\boldsymbol{\theta}}(\boldsymbol{a})(\boldsymbol{y}) = \left\langle \boldsymbol{H}_a^{(L)}, \boldsymbol{H}_y^{(L)} \right\rangle, \tag{L.20}$$

where $\odot$ denotes point-wise multiplication, $\phi$ denotes a activation function, and $\boldsymbol{\theta}$ represents all trainable parameters of the DeepONet model. In particular, $\{\boldsymbol{W}_a^{(l)}, \boldsymbol{b}_a^{(l+1)}\}_{l=1}^{L+1}$ and $\{\boldsymbol{W}_y^{(l)}, \boldsymbol{b}_y^{(l+1)}\}_{l=1}^{L+1}$ are the weights and biases of the branch and trunk networks, respectively. we embed the DeepONet inputs $\boldsymbol{a}$ and $\boldsymbol{y}$ into a high-dimensional feature space via two encoders $\boldsymbol{U}, \boldsymbol{V}$, respectively. Instead of just merging the propagated information in the output layer of the branch and trunk networks, we merge the embeddings $\boldsymbol{U}, \boldsymbol{V}$ in each hidden layer of these two sub-networks using a point-wise multiplication (equation (L.17) - (L.18)). Heuristically, this design may not only help input signals propagate through the DeepONet, but also enhance its capability of representing non-linearity due to the extensive use of point-wise multiplications.

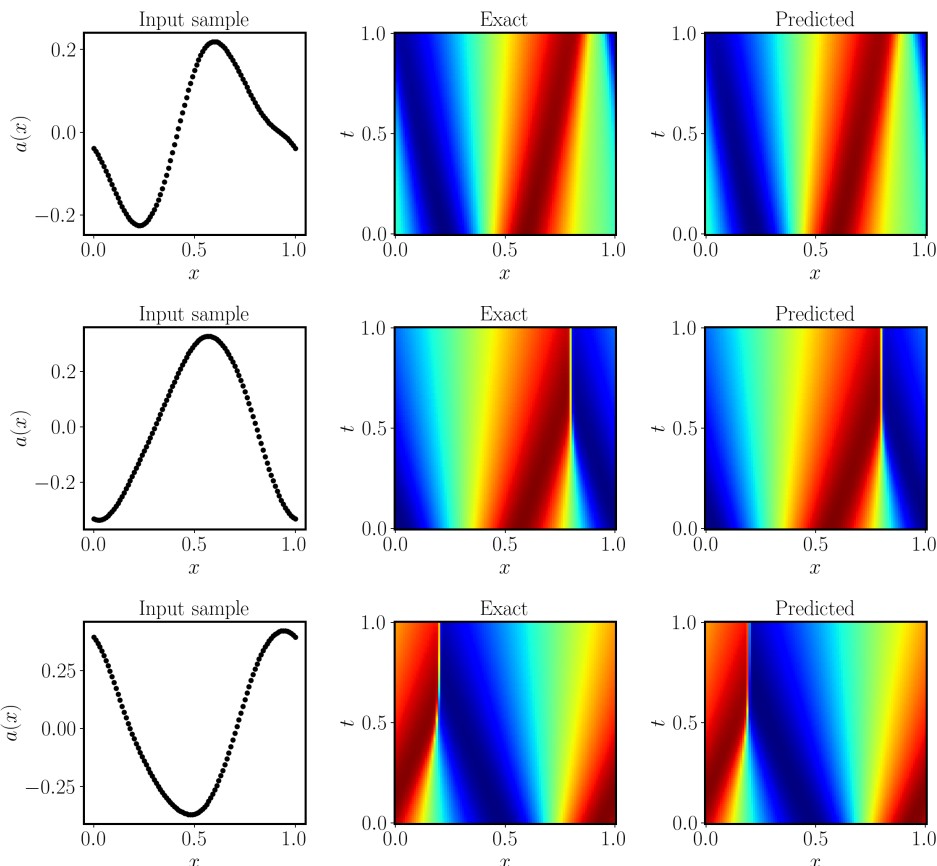

Figure 25: *Burgers PDE:* Predicted solutions of a trained physics-informed DeepONet with random weight factorization, corresponding to randomly chosen input samples in the test data-set.

