# OpenReview forum: "Random Weight Factorization improves the training of Continuous Neural Representations"
_ICLR.cc/2023/Conference — Submitted to ICLR 2023_

### Official Review · Reviewer_AGKQ · 2022-10-16

**Confidence:** 4
**Correctness:** 4
**Technical Novelty And Significance:** 3
**Empirical Novelty And Significance:** 3
**Recommendation:** 8

**Clarity, Quality, Novelty And Reproducibility:**

The paper is quite clear, and the theoretical and experimental results are of high quality and justify the claims made in the paper. The proposed method is related to many existing methods and is simple, but I find it to be useful and the benefits are justified very well and in a new way, resulting in a novel contribution. I believe the result is reproducible.

**Strength And Weaknesses:**

In my opinion the strengths of the paper are:
- The paper is extremely clear in description of the proposed method and exposition of theoretical and experimental results. This is a plus, as it will make the impact of the paper significantly higher since it can easily be adapted into application pipelines.
- The claims of improvement are backed up both theoretically and experimentally, and seem significantly better than the comparable baseline methods. The breadth of experimental experiments for which coordinate-based networks are used is great, and I find it valuable that in all of these situations the randomized weight factorization provides the same improvement.

In my opinion, there are no glaring weaknesses in the paper. One thing which would be interesting to be directly discussed would be the aspect of computational efficiency. Specifically, does the factorization increase the size of the models, since there are more parameters being optimized after the factorization. Additionally, does the improvement in loss landscape significantly improve the training time of coordinate-based networks across all applications. This is an important question for those who work in neural rendering (ex: with neural radiance fields), and this could be a simple but applicable solution for this.


**Summary Of The Paper:**

The paper proposes a method for improving and accelerating the training of coordinate-based networks by proposing randomized weight factorization. This factorization generalizes weight normalization, but can be applied in the coordinate-based network setting for signal memorization. It is done by factoring each of the weights in the layers into a multiplication between a diagonal scale matrix and a dense weight matrix. The paper demonstrates that this factorization theoretically improves the distance to a local minima, resulting in easier optimization and robustness to poor initializations. This is experimentally validated across a number of signal fitting tasks which coordinate-based networks are applied to, including image and 3D shape memorization, neural radiance field optimization, and partial differential equation solutions.

**Summary Of The Review:**

This paper proposes a new factorization for fully connected coordinate-based network architectures, theoretically proves that it has some properties which are amenable to better initialization (resulting in better results), and experimentally validates that this is the case. I find this to be a convincing story, and this is a topic which is very interesting for those who work on applications involving coordinate-based networks. Thus, I believe this paper would have a high impact and recommend it for acceptance.

POST REBUTTAL UPDATE: After reviewing the authors' response, and the other reviews and responses, I am not inclined to change my score. The authors have empirically addressed the hyperparameter comment, which I believe was valid, and have shown that their factorization leads to improvements. The theoretical section has been improved with respect to prior work. I believe the improvements are noticeable, and I don't believe comments on vastly different architectures are relevant since the paper specifically focuses on coordinate-based networks.

---

> ### Author Response · Authors · 2022-11-16
> **Response to Reviewer AGKQ**
>
> We thank the reviewer for the detailed feedback and are happy to receive the positive assessment of our contribution, presentation, and empirical evaluations. Below we address the questions; we have also updated the manuscript to accommodate the feedback.
>
> ***Q1: One thing which would be interesting to be directly discussed would be the aspect of computational efficiency. Specifically, does the factorization increase the size of the models, since there are more parameters being optimized after the factorization. Additionally, does the improvement in loss landscape significantly improve the training time of coordinate-based networks across all applications.***
>
> The factorization moderately increases the size of the models by $\mathcal{O}(LD)$, where $L$ denotes the number of layers and $D$ the width of the network. It is worth noting that the computational overhead of our method is marginal. This is further clarified by Table 7 where we report the computational cost in terms of training iterations per second for the  networks employed in each benchmark.

---

> > ### Comment · Reviewer_AGKQ · 2022-12-02
> > **Response**
> >
> > Thank you for the response, I do not have any further questions. I have updated my review to reflect my opinions about the paper after all other responses.

---

> > > ### Author Response · Authors · 2022-12-07
> > > **Thank you!**
> > >
> > > Dear Reviewer,
> > >
> > > We sincerely appreciate your positive feedback, the time and effort you have put into reviewing our modifications and responses! We are happy to answer any potential follow-up questions.
> > >
> > > Best,
> > >
> > > Authors

---

### Official Review · Reviewer_chW4 · 2022-10-22

**Confidence:** 3
**Correctness:** 3
**Technical Novelty And Significance:** 3
**Empirical Novelty And Significance:** 3
**Recommendation:** 6

**Clarity, Quality, Novelty And Reproducibility:**

The paper is quite clear overall, could be improved by:
- giving the definition of the "distance" measure in Theorem 1 explicitly, and
- having a formal algorithm in the main text and explaining why authors propose using $\exp(s)$ instead of $s$.

Regarding quality and novelty, I do not think any more work is needed.

About the reproducibility, having codes publicly available will be helpful.

**Strength And Weaknesses:**

__Strength__
- The method is very simple and easy to use, can be used as a drop-in replacement of the weight matrices of INR. I especially like the fact that the method may not incur too much additional compute / memory on top of the vanilla INR training procedure, as the training cost is one of the central issues of INR.

__Weaknesses__
- The "exponentiation" in the main algorithm (in appendix B) is not well-motivated, and is not well discussed in the main text. Could authors give a little more explanation why using $\exp(s)$ instead of $s$ is a good thing to do? Perhaps a formal ablation study or theoretical justification may help.

- The explanation regarding SIREN is not clear enough to me. In page 4, authors argue that (1) using RWP is better than using the SIREN (2) the benefit of SIREN is mostly from using the reparameterization $\omega_0 \cdot \mathbf{w}$ rather than the periodic activation. I am not really sure about how authors showed this in Appendix G and H. It seems like in Appendix G and H, authors try various values of $\omega_0$ of SIREN to show that the peak performance of such fixed parameterization is lower than the value achieved by RWP. But it is not clear from the text with which activation function RWP and fixed $\omega$ are trained with. Did you use periodic activation function, or the ReLU? In either case, I do not see how this experiment leads to a conclusion that having periodic activation is less important than having a scaling factor in SIREN.

- Related to the previous point, I think that perhaps showing that RWP reparameterization boosts the performance of INRs even when the underlying model architecture uses a periodic activation function (like SIREN) is a right way to prove the benefit of RWP. Indeed, in many cases, SIREN is known to achieve way better/faster convergence than the ReLU-based models used in the experiments.

- In fact, I wanted to test the claims of the authors by myself but the code has not been provided (unfortunately). Providing codes will help readers validate the claim.

- I wonder if such reparameterization will help boosting the performance of meta-learned INRs (via MAML or hypernetworks). Is there any related experiment?

- In Theorem 1, I am not sure how meaningful having a small "distance" (which has not been properly defined, by the way) is, because the reparameterization also have a critical impact on the scale of the gradient of each component, s and V. In other words, despite having a smaller distance by the reparameterization, the effective number of optimization steps can be similar to the original parameterization.

- Results like Theorem 2 has already appeared in the work of Arora et al. (2018), which should be mentioned in the text. In fact, the theoretical results of Arora et al. may help authors explain the good performance of the proposed method.

---
Arora et al., "On the optimization of deep networks: Implicit acceleration by overparameterization," ICML 2018.

**Summary Of The Paper:**

This paper proposes a simple but well-performing technique of reparameterizing each weight matrix of implicit neural representations (INRs) as a product of diagonal matrix (with diagonal elements being equal to the exponential of a single parameter) and another matrix. Empirically, it turns out that the reparameterized INR converges to a better-fitting solution to the signal being trained.

**Summary Of The Review:**

The proposed method is potentially a very useful technique for training implicit neural representations, but I have several concerns regarding the empirical validations of the technique, especially on how the method can be combined with the state-of-the-art architecture, SIREN.

---

> ### Author Response · Authors · 2022-11-16
> **Response to Reviewer chW4 (Part I)**
>
> We thank the reviewer for providing valuable feedback to improve the work. We have carefully addressed them and have made the necessary changes accordingly, as detailed below.
>
> ***Q1: The "exponentiation" in the main algorithm (in appendix B) is not well-motivated, and is not well discussed in the main text. Could authors give a little more explanation why using $\exp(\mathbf{s})$ instead of $ \mathbf{s}$ is a good thing to do? Perhaps a formal ablation study or theoretical justification may help.***
>
> We apologize for insufficient clarity on this point.  The main purpose of the exponentiation is to strictly avoid zeros or very small values in the scale factor, as our method requires to initialize $\mathbf{v}$ by $\mathbf{v} = \mathbf{w} / s$.   Moreover, this  exponential parameterization is motivated by the original work of Weight Normalization [1] where the authors proposed to use $\exp(s)$ instead of working with $s$ directly. Parameterizing the scale factor in the log-scale more easily allows it to span a wide range of different magnitudes.
>
> Furthermore, for the 1D regression benchmark, we have performed an additional ablation study to quantify the effect of different distributions (Normal, LogNormal,Uniform, LogUniform) for initializing $\mathbf{s}$  and report the average and standard deviation of mean square errors over 10 random seeds. To facilitate a fair comparison, we manually tune each distribution such that the sampled values are roughly in the same range. As shown in the table below, we see that LogNormal is slightly better than the other distributions while overall there is no significant discrepancies in terms of the accuracy for using different distributions.
>
> | Distribution |           MSE           |
> |:------------:|:-----------------------:|
> |    Normal    | $3.95e-04 \pm 2.22e-04$ |
> |   LogNormal  | $3.76e-04 \pm 3.31e-04$ |
> |    Uniform   | $3.94e-04 \pm 4.89e-04$ |
> |  LogUniform  | $5.40e-04 \pm 4.78e-04$ |
>
> ***Q2: The explanation regarding SIREN is not clear enough to me. In page 4, authors argue that (1) using RWP is better than using the SIREN (2) the benefit of SIREN is mostly from using the reparameterization $\omega_0 \cdot w$ rather than the periodic activation. I am not really sure about how authors showed this in Appendix G and H. It seems like in Appendix G and H, authors try various values of $\omega_0$ of SIREN to show that the peak performance of such fixed parameterization is lower than the value achieved by RWP. But it is not clear from the text with which activation function RWP and fixed $\omega_0$  are trained with. Did you use periodic activation function, or the ReLU? In either case, I do not see how this experiment leads to a conclusion that having periodic activation is less important than having a scaling factor in SIREN.***
>
> We apologize for the confusion. In the experiments, we compare the performance of SIREN against our baseline, i.e. a plain MLP equipped with ReLU activations and no input mappings, under exactly the same hyper-parameter settings. Here we emphasize that the plain MLP is initialized using the conventional Glorot scheme and does not employ  random weight factorization. As a result, for $w_0 = 1$ in SIREN, the main difference between SIREN and our baseline is the activation function (sine vs. ReLU). However, in Figure 11 and Figure 16, it can be observed that when $w_0 = 1$, SIREN achieves a similar performance as our baseline. This suggests that the periodic activation function alone does not play a crucial role in improving the model performance. In the same figures, we can see that the SIREN performance is mostly affected by the value of $w_0$. Therefore, we specifically attribute the success of SIREN to the choice of the scale factor $w_0$. Indeed, the use of periodic activations in SIREN mainly serves to provide a smooth neural representation of complex natural signals and their derivatives.
>
> [1] Tim Salimans and Durk P Kingma. Weight normalization: A simple reparameterization to accelerate training of deep neural networks. Advances in neural information processing systems, 29, 2016.

---

> > ### Author Response · Authors · 2022-11-16
> > **Response to Reviewer chW4 (Part II)**
> >
> > ***Q3: Related to the previous point, I think that perhaps showing that RWP reparameterization boosts the performance of INRs even when the underlying model architecture uses a periodic activation function (like SIREN) is a right way to prove the benefit of RWP. Indeed, in many cases, SIREN is known to achieve way better and faster convergence than the ReLU-based models used in the experiments***
> >
> > This is an excellent suggestion. In conjunction with the comments of reviewer C8fw, we have conducted a comprehensive hyper-parameter sweep over  different random seeds, learning rates, activation functions, scales of the Fourier features, and the width and depth of the network. In the table below, we fix the sine activation function and report the averaged PSNR over the rest of hyper-parameters. The results suggest that the RWF can also benefit MLPs equipped with periodic activations, outperforming the other competing approaches.  Additional visualizations are provided in Figures 7,8,12,13 in the revised Appendix.
> >
> >
> > | Dataset |      Plain   |        AA        |        WN        |          RWF         |
> > |:-------:|:----------------:|:----------------:|:----------------:|:--------------------:|
> > | Natural | $19.90 \pm 2.11$ | $20.12 \pm 2.03$ | $19.96 \pm 2.17$ | **21.39 $\pm$ 1.40** |
> > |   Text  | $24.86 \pm 4.56$ | $25.55 \pm 4.64$ | $25.15 \pm 4.41$ | **27.07 $\pm$ 4.18** |
> > |  Shepp  | $23.91 \pm 3.12$ | $23.97 \pm 3.22$ | $24.25 \pm 3.05$ | **24.89 $\pm$ 3.60** |
> > |  Atlas  | $19.13 \pm 2.67$ | $19.27 \pm 2.60$ | $19.46 \pm 2.66$ | **20.13 $\pm$ 2.41** |
> >
> >
> > ***Q4: In fact, I wanted to test the claims of the authors by myself but the code has not been provided (unfortunately). Providing codes will help readers validate the claim.***
> >
> > As suggested by the reviewer, we have uploaded some demos in the supplementary material for validation purposes. To preserve anonymity we plan to release code for reproducing all the results reported here after the review process.
> >
> > ***Q5: I wonder if such reparameterization will help boosting the performance of meta-learned INRs (via MAML or hypernetworks). Is there any related experiment?***
> >
> > This is indeed a very interesting question and opens new venues for the future work.  We believe that the random weight factorization can potentially benefit meta-learned INRs. Unfortunately, we have not conducted the related experiments so far due to the tight deadline of rebuttal. We would like to investigate these interesting topics in the future.
> >
> > ***Q6: In Theorem 1, I am not sure how meaningful having a small "distance" (which has not been properly defined, by the way) is, because the reparameterization also have a critical impact on the scale of the gradient of each component, s and V. In other words, despite having a smaller distance by the reparameterization, the effective number of optimization steps can be similar to the original parameterization.***
> >
> > We agree with the reviewer's assessment. It is true that a very small distance may not be beneficial for  training. As illustrated in Figure 6, we can see that a large mean for the scale factors can lead to degraded performance. This observation motivates to select an appropriate distribution to initialize the scale factors. From our experience, a moderate range generated by $\mathbf{s}\sim\exp(\mathcal{N}(1, 0.1))$ typically yields good performance. Alternatively, one can always fine-tune the distribution of $\mathbf{s}$ at initialization by performing a hyper-parameter sweep over a validation date-set.
> >
> > ***Q7: Results like Theorem 2 has already appeared in the work of Arora et al. (2018), which should be mentioned in the text. In fact, the theoretical results of Arora et al. may help authors explain the good performance of the proposed method.***
> >
> > We thank the reviewer for pointing this out. We have cited this work and revised our manuscript to acknowledge its contribution.
> >
> > ***Q8: The paper is quite clear overall, could be improved by:***
> > -  ***giving the definition of the "distance" measure in Theorem 1 explicitly***
> > -  ***having a formal algorithm in the main text and explaining why authors propose using $\exp(s)$  instead of $s$.***
> >
> > We appreciate the reviewer's suggestions. The distance rigorously is defined as
> >
> > $\text{dist}(A, B) = \min_{x \in A, y \in B} \|x - y\| $
> >
> > where $A, B$ are two sets in $\mathbb{R}^n$, and $\|\cdot \|$ denotes the $L^2$ norm. We have also updated it in the main text.
> >
> > According to our previous response, we have added a detailed explanation of using $\exp(s)$ in the main text and an additional ablation study in Appendix. Unfortunately, we apologize that we may not be able to present a formal algorithm in the main text because it exceeds the limit of 9 pages. Instead, we have included the algorithm and implementation details in Appendices B and C.

---

> > > ### Comment · Reviewer_chW4 · 2022-12-02
> > > **Thank you for the response.**
> > >
> > > Dear authors,
> > >
> > > thank you for the response. Raised a score.

---

> > > > ### Author Response · Authors · 2022-12-07
> > > > **Thank you!**
> > > >
> > > > Dear Reviewer,
> > > >
> > > > We sincerely appreciate your positive feedback, the time and effort you have put into reviewing our modifications and responses! We are happy to answer any potential follow-up questions.
> > > >
> > > > Best,
> > > >
> > > > Authors

---

### Official Review · Reviewer_mJaA · 2022-10-23

**Confidence:** 4
**Clarity, Quality, Novelty And Reproducibility:** Clarity, Quality, and Reproducibility…
**Correctness:** 4
**Technical Novelty And Significance:** 2
**Empirical Novelty And Significance:** 2
**Recommendation:** 5

**Strength And Weaknesses:**

Strength:
Overall, the paper is well-organized and easy to follow. This paper studies how to better initialize and factorize the weight parameters of the multi-layer perception. Specifically, they suggest using random weight factorization to factorize the original weight parameters. They first theoretically show that the space of this factorization is flexible enough to be arbitrarily close to the global minima. After that, they show that the proposed weight factorization can be recognized as a gradient descent with a self-adaptive learning rate. I also admire the tremendous experimental work in this paper.
Weakness:
1. My biggest concern is that I didn’t find much difference between the proposed method and the adaptive activation function method. I check the algorithm in Appendix B. It seems that the random weight factorization is only applied once in the initialization. After that, the model was trained with the standard gradient descent method. If we omit the initialization step, this is almost like training a model with an adaptive activation function. I am also concerned about how to ensure that the initialized scale factor s^{(l)} is better than the vector of ones. Please correct me if I am wrong.

2. The author claims that the proposed method can effectively mitigate spectral bias. But I am not aware of any such analysis in the experiment part.


**Summary Of The Paper:**

This work proposes random weight factorization as an initialization of the conventional linear layers. Theoretically, it is shown how this factorization alters the underlying loss landscape and effectively enables each neuron in the network to learn using its own self-adaptive learning rate. Empirically, the proposed initialization method can result in improving the training on a variety of tasks, including image regression, shape representation, computed tomography, inverse rendering, solving partial differential equations, and learning operators between function spaces.

**Summary Of The Review:**

I didn’t find much difference in this work compared to the adaptive activation function method. In addition, the random weight factorization method only applies to the initialization.

---

> ### Author Response · Authors · 2022-11-16
> **Response to Reviewer mJaA**
>
> We would like to thank you for your time spent reviewing our paper and providing constructive comments. Please kindly find our responses to your raised questions below.
>
> ***Q1: My biggest concern is that I didn’t find much difference between the proposed method and the adaptive activation function method. I check the algorithm in Appendix B. It seems that the random weight factorization is only applied once in the initialization. After that, the model was trained with the standard gradient descent method. If we omit the initialization step, this is almost like training a model with an adaptive activation function. I am also concerned about how to ensure that the initialized scale factor s^{(l)} is better than the vector of ones. Please correct me if I am wrong.***
>
> We agree with the referee that the proposed method bears some similarity to the adaptive activation method of Jagtap *et. al.* [1]. During training, both approaches have a similar forward propagation rule and optimize the scale factors and weight matrix of each layer with gradient descent. However, we argue that the key difference between these two methods is the random initialization step which should be neither overlooked or omitted.
>
> At initialization, the adaptive activation approach typically initializes the scale factor by a vector of ones and the weight matrices by a standard initialization scheme (e.g. Glorot). This yields a trivial weight factorization $\mathbf{w} = 1 \cdot \mathbf{w}$. In contrast, the proposed random weight factorization initializes the scale factors $\mathbf{s}$ by a random distribution and re-parameterizes the initialized weight matrices accordingly by $\mathbf{w} = e^{\mathbf{s}} \cdot \mathbf{v}$.
>
> The proposed initialization of the scale factors is directly motivated by the geometric intuition provided in Figure 1 and Theorem 1, which show that the distance between factorizations representing the initial parameters and the global minimum becomes smaller in the factorized parameter space for larger values of $\mathbf{s}$. We therefore initialize the scale factor $\exp(s)$ with a distribution that concentrates around values moderately greater than 1 to reduce the distance between the initialization and a proper local minimum, therefore speeding up the training process.  Our experiments additionally give strong support to this claim: the random weight factorization yields consistent and substantial improvements across a wide range of applications, while the adaptive activation generally performs similar to a plain MLP baseline.
>
> ***Q2: The author claims that the proposed method can effectively mitigate spectral bias. But I am not aware of any such analysis in the experiment part.***
>
> As demonstrated in [2] and [3], the eigenvalues of the Neural Tangent Kernel (NTK) characterize the convergence rate of the corresponding kernel eigenvectors. Specifically, components of the target function that are aligned with kernel eigenvectors corresponding to larger eigenvalues will be learned faster. Therefore, the rapid decay of the NTK eigenvalues implies extremely slow convergence to  high frequency components of the target function, a pathology commonly referred to as ``spectral bias''.
>
> In the reported experiments for 1D regression, we have computed the eigenvalues of the resulting empirical Neural Tangent Kernel (NTK) at the last step of training, as shown in Figure 2. We can observe that the proposed method (RWF) leads to a flatter NTK spectrum and slower eigenvalue decay than the competing approaches, indicating better-conditioned training dynamics and less severe spectral bias.
>
>
> [1] Ameya D Jagtap, Kenji Kawaguchi, and George Em Karniadakis. Adaptive activation functions accelerate convergence in deep and physics-informed neural networks. Journal of Computational Physics, 404:109136, 2020.
>
> [2] Matthew Tancik, Pratul Srinivasan, Ben Mildenhall, Sara Fridovich-Keil, Nithin Raghavan, Utkarsh Singhal, Ravi Ramamoorthi, Jonathan Barron, and Ren Ng. Fourier features let networks learn high frequency functions in low dimensional domains. Advances in Neural Information Processing Systems, 33:7537–7547, 2020.
>
> [3] Sifan Wang, Hanwen Wang, and Paris Perdikaris. On the eigenvector bias of fourier feature networks: From regression to solving multi-scale PDEs with physics-informed neural networks. Computer Methods in Applied Mechanics and Engineering, 384:113938, 2021.

---

> > ### Comment · Reviewer_mJaA · 2022-12-07
> > **Response**
> >
> > Thank you for the response. I am satisfied with the author's response to Q2. However, for Q1, I don't believe the random weight factorization can reduce the distance between the initialization and a proper local minimum. At least, I didn't see a way to prove this statement theoretically despite the tremendous empirical advantage of this method. I have raised a score.

---

> > > ### Author Response · Authors · 2022-12-07
> > > **weight factorization can reduce the distance between the initialization and a proper local minimum**
> > >
> > > We are happy to see that the referee’s initial concerns about the novelty of our approach have now been rectified.
> > >
> > > Regarding the follow up comment on the effect of random weight factorization on reducing the distance between an initialization and a proper local minimum, we would like to clarify that this refers to the distance between the ''hyperbolas'' in the factorized parameter space, as schematically illustrated in Figure 1 (right panel). Indeed, in this setting it is straightforward to prove that the distance between any two ''hyperbolas'' (e.g. corresponding to different initializations and/or local minima) in the factorized parameter space decreases, as the scale factors $\mathbf{s}$ are increased. The theoretical proof can be constructed as follows.
> > >
> > > Starting from any fixed network parameters  $\mathbf{\theta} = \\{\mathbf{W}^{(l)}, \mathbf{b}^{(l)}\\}_{l=1}^{L+1}$, we consider the following weight factorization
> > >
> > > $ \text{diag}( \mathbf{s}^{(l)})  \cdot \mathbf{V}^{(l)} = \mathbf{W}^{(l)}, \quad l=1, 2, \dots, L+1. $
> > >
> > > Next, consider the set of all possible weight factorizations associated with the initialization $\mathbf{\theta}$ as
> > >
> > > $ U_{\mathbf{\theta}} = \\{ (\mathbf{s}^{(l)}, \mathbf{V}^{(l)})_{l=1}^{L+1}  : \text{diag}(\mathbf{s}^{(l)}) \cdot  \mathbf{V}^{(l)}   = \mathbf{W}^{(l)}, \quad l=1, \dots, L+1  \\}. $
> > >
> > > Let us now define $U_0$ in the factorized parameter space by
> > >
> > > $U_0 = \\{( \mathbf{s}^{(l)}, \mathbf{0})_{l=1}^{L+1} : \mathbf{s}^{(l)} \in \mathbb{R}^{d_l}, \quad l=1,\dots, L+1 \\}. $
> > >
> > > Since $\text{diag}(\mathbf{s}^{(l)}) \cdot  \mathbf{V}^{(l)}   = \mathbf{W}^{(l)}$ and the network parameters $\mathbf{\theta}$ are fixed, there exists a constant $C(\theta)$  such that
> > > $ \\|\mathbf{V}^{(l)}\\| \leq \frac{\\|\mathbf{W}^{(l)}\\|}{\\|\mathbf{s}^{(l)}\\|} \leq \frac{C(\theta)}{\\|\mathbf{s}^{(l)}\\|},  \quad l=1, \dots, L+1. $
> > >
> > > Then, for any weight factorization $(\mathbf{s}^{(l)}, \mathbf{V}^{(l)})_{l=1}^{L+1}$,  we can take
> > >
> > > $(\mathbf{s}^{(l)}, \mathbf{0})_{l=1}^{L+1} \in U_0$.
> > >
> > > By the definition of distance between sets, we obtain
> > >
> > > $\text{dist}(U_{\mathbf{\theta}}, U_0) = \min_{\mathbf{x} \in U_{\mathbf{\theta}}, \mathbf{y} \in  U_0} \\|\mathbf{x} - \mathbf{y}\\|
> > >      \leq \sqrt{\sum_{l=1}^{L+1} \\|\mathbf{V}^{(l)} \\|^2 }  \leq C(\theta) \sqrt{\sum_{l=1}^{L+1} \frac{1}{\\|\mathbf{s}^{(l)}\\|^2}  }. $
> > >
> > > Therefore, for any network parameters $\theta, \theta'$, taking $C=\max\\{C(\theta), C(\theta')\\}$ yields
> > >
> > > $ \text{dist}(U_{\mathbf{\theta}}, U_{\mathbf{\theta}'}) \leq \text{dist}(U_{\mathbf{\theta}}, U_0) + \text{dist}(U_0, U_{\mathbf{\theta}'}) \leq 2C\sqrt{\sum_{l=1}^{L+1} \frac{1}{\\|\mathbf{s}^{(l)}\\|^2}  }. $
> > >
> > > As a corollary, let $\theta$ denote a network initialization and $\theta_*$ be a proper local minimum, then there exists a weight factorization with large enough scale factors $\mathbf{s}$,
> > > such that the distance between $\theta$ and $\theta_*$ can be arbitrarily small in the factorized parameter space.
> > >
> > > While we acknowledge that it is hard to explicitly quantify the implications of this result on the practical training of continuous neural representations, our main goal here is to provide an intuitive explanation for a possible mechanism behind the inner workings of random weight factorization, the effectiveness of which has been extensively verified in our numerical experiments.

---

> > > > ### Author Response · Authors · 2022-12-11
> > > > **Discussion period ending soon**
> > > >
> > > > Dear Reviewer mJaA,
> > > >
> > > > We would like to thank the reviewer again for your time and effort.
> > > >
> > > > Given there is only ***1 days left*** in the discussion period, we wanted to double check if the reviewer had seen our latest comment and whether there were any final clarifications the reviewer would like. If the concerns of the reviewer are clarified and the reviewer is convinced of the novelty and contribution of our work, we sincerely appreciate it if the reviewer could update your review and score to reflect that.
> > > >
> > > > Once again, many thanks for your time and dedication to the review process, we are extremely grateful.
> > > >
> > > > Best,
> > > >
> > > > Authors

---

### Official Review · Reviewer_C8fw · 2022-10-24

**Confidence:** 3
**Correctness:** 3
**Technical Novelty And Significance:** 2
**Empirical Novelty And Significance:** 2
**Recommendation:** 5

**Clarity, Quality, Novelty And Reproducibility:**

**Clarity** Good, but given how simple the method is, this is to be expected.

**Quality** I think the experimental setup is flawed and

**Novelty** New as far as I'm aware, although highly similar to weight normalization, but this is acknowledged and discussed explicitly. The parametrization is not particularly involved, but discovering small variations on something as basic as linear layer parametrization is not trivial.

**Reproducibility** Hyperparameters seem to be reported exhaustively in the appendix. However, I could not find any discussion of how they were obtained, so while the experiments may be re-runnable, the overall process of the study is not reproducible.

**Strength And Weaknesses:**

Strengths:
+ The proposed method is very simple.
+ The method is scalable and potentially widely applicable.
+ The paper presents results on a broad range of settings for the experiments.

Weaknesses:
- My understanding based on Tab. 3 in the appendix is that the optimizer setup is fixed across the parametrizations that are compared in each benchmark. As I read it, the paper argues that random weight factorization outperforms the alternatives, i.e. the best case performance is better. However, then the optimization setup would need to be tuned for each method independently. As it stands, the paper can roughly support the statement "For a range of benchmarks there exists some setting of the hyperparameter where the proposed method performs best". However, this is not all too relevant, it would either need to typically perform best across a range of hyperparameter setups (i.e. be more robust) or consistently deliver the best optimal performance. To me, the experimental setup is critically flawed and would need to be addressed for me to recommend acceptance (I'm open to discussion on this point and do not strictly expect all experiments to be rerun with a hyperparameter sweep for each baseline).
- There are no error bars in most experiments and I did not find any mention of multiple runs with different random seeds, which significantly weakens any reported performance gains. Also as a result, I'm not sure how to interpret e.g. the ablation study in Fig 6, it seems extremely noisy across the board with no observable trend beyond high errors for a mean of 4.
- The reported performance gains are mostly incremental (although this is probably not unexpected with a relatively small change such as the weight parametrization on something as basic as feedforward neural networks with best practices for training them that have been established over the past decade).
- The range of architectures is quite narrow, it seems like the parametrization would be straight-forward to plug into CNNs and Transformers if I'm not missing anything.
- I don't find the argument around the proximity of parametrizations too convincing, I'd expect these to be increasingly far away from the origin, especially in high dimensions. This probably would make them hard to reach by gradient-based optimization, in particular in conjunction with L2 regularization (although I suppose it depends whether that is applied to the weight matrices or the actual parameters). I would appreciate some more discussion around this or some small experiment on an actual neural network architecture.

Other:
* I haven't seen any of the benchmarks before, so find it difficult to judge the reported performance gains.

**Summary Of The Paper:**

This paper proposes random weight factorization, a new parametrization for linear neural network layers inspired by weight normalization. It argues that the parametrization reduces the distance between different parameter configurations in the loss landscape and reduces spectral bias, leading to performance improvements across a range of tasks.

**Summary Of The Review:**

This is a simple, but potentially broadly useful idea. To me this puts the bar on the empirical evaluation rather high. Unfortunately, I believe that the evaluation falls short of that, so I do not recommend acceptance.

---

> ### Author Response · Authors · 2022-11-16
> **Response to Reviewer C8fw (Part I)**
>
> We would like to thank the reviewer for the positive feedback and constructive comments. We have carefully addressed them and have made the necessary changes accordingly, as detailed below.
>
> ***Q1: My understanding based on Tab. 3 in the appendix is that the optimizer setup is fixed across the parametrizations that are compared in each benchmark. As I read it, the paper argues that random weight factorization outperforms the alternatives, i.e. the best case performance is better. However, then the optimization setup would need to be tuned for each method independently. As it stands, the paper can roughly support the statement "For a range of benchmarks there exists some setting of the hyperparameter where the proposed method performs best". However, this is not all too relevant, it would either need to typically perform best across a range of hyperparameter setups (i.e. be more robust) or consistently deliver the best optimal performance. To me, the experimental setup is critically flawed and would need to be addressed for me to recommend acceptance (I'm open to discussion on this point and do not strictly expect all experiments to be rerun with a hyperparameter sweep for each baseline).***
>
> The referee is raising a fair point which we hereby address by performing a comprehensive study of hyper-parameter sweeps across different model configurations and benchmarks, resulting in a total of more than  10,000 additional numerical experiments. Specifically, we revisit the Image Regression and Computed Tomography benchmarks and train models with different weight parametrizations (Plain, Adaptive Activation, Weight Normalization, Random Weight Factorization) over different random seeds, learning rates, activation functions, scale hyper-parameters, as well as MLP architectures with different depths and widths:
>
> | Hyper-parameters |      Values      |
> |:----------------:|:----------------:|
> |    Random Seed   |  2, 3, 5, 7, 11  |
> |   Learning rate  |    1e-3, 1e-4    |
> |    Activation    | ReLU, GELU, Sine |
> |  Gaussian scales |     1, 5, 10     |
> |       Width      |   64, 128, 256   |
> |       Depth      |      3, 4, 5     |
>
> Our main observations here are consistent with the original claims discussed in our paper, and can be summarized as
>
> - Over the entire range of hyper-parameter settings tested, random weight factorization (RWF) consistently yields the best optimal performance compared to the alternatives, as claimed in Table 1 of the main text.
>
> - Additionally, averaged over all hyperparameter settings, we observe that random weight factorization (RWF) on average outperforms the competing approaches (see Table below).
>
> |         Task        |       Metric      |   Case  |       Plain      |        AA        |        WN        |          RWF          |
> |:-------------------:|:-----------------:|:-------:|:----------------:|:----------------:|:----------------:|:---------------------:|
> |   Image Regression  | PSNR ($\uparrow$) | Natural | $19.85 \pm 2.00$ | $20.01 \pm 1.96$ | $19.82 \pm 2.01$ | **21.12 $\pm$ 1.62** |
> |   Image Regression  | PSNR ($\uparrow$) |   Text  | $24.80 \pm 4.85$ | $25.16 \pm 4.87$ | $24.90 \pm 4.82$ | **26.71 $\pm$ 4.45** |
> | Computed Tomography | PSNR ($\uparrow$) |  Shepp  | $24.95 \pm 3.37$ | $25.09 \pm 3.49$ | $25.24 \pm 3.14$ |  **26.81 $\pm$ 3.81** |
> | Computed Tomography | PSNR ($\uparrow$) |  Atlas  | $19.37 \pm 2.78$ | $19.47 \pm 2.73$ | $19.66 \pm 2.81$ | **20.51 $\pm$ 2.55** |
>
> We have provided additional figures (Figures 7,8,12,13 in the Appendix) and revised the paper accordingly. A detailed log of the aforementioned experiments that support our conclusions can be accessed in the supplementary material.

---

> > ### Author Response · Authors · 2022-11-16
> > **Response to Reviewer C8fw (Part II)**
> >
> > ***Q2: There are no error bars in most experiments and I did not find any mention of multiple runs with different random seeds, which significantly weakens any reported performance gains. Also as a result, I'm not sure how to interpret e.g. the ablation study in Fig 6, it seems extremely noisy across the board with no observable trend beyond high errors for a mean of 4.***
> >
> > We thank the reviewer for raising this point. For the task of inverse rendering and solving PDEs, we repeat each experiment with 10 different random seeds and compute the mean and standard deviation of each corresponding performance metric  across seeds (see Table below). We reach the same conclusion, namely that the proposed random weight factorization yields the best performance and have updated Table 1 accordingly.
> >
> > |        Task       |           Metric          |      Case     |         Plain        |           AA          |          WN          |              RWF              |
> > |:-----------------:|:-------------------------:|:-------------:|:--------------------:|:---------------------:|:--------------------:|:-----------------------------:|
> > | Inverse Rendering |     PSNR ($\uparrow$)     |      Lego     |   $26.11 \pm 0.06$   |    $26.11 \pm 0.07$   |   $26.07 \pm 0.07$   |   **26.21 $\pm$ 0.05**   |
> > |    Solving PDEs   | Rel. $L^2$ ($\downarrow$) |   Advection   | 25.94% $\pm$ 8.18% | 36.27% $\pm$ 16.12% | 67.40% $\pm$ 1.72% | **2.26% $\pm$ 0.63%** |
> > |    Solving PDEs   | Rel. $L^2$ ($\downarrow$) | Navier-Stokes | 11.93% $\pm$ 4.59% |  10.95% $\pm$ 2.76% | 14.62% $\pm$ 4.04% | **6.79% $\pm$ 1.25%** |
> >
> > We have also conducted the same ablation experiment over a wider range of mean $\mu$ and 10 random seeds, and updated Figure 6 in the manuscript. One can observe that all the curves are roughly basin-shaped, indicating a ``sweet spot'' for the random weight factorization, i.e., $\mu \in [1,2]$.

---

> > > ### Author Response · Authors · 2022-11-16
> > > **Response to Reviewer C8fw (Part III)**
> > >
> > > ***Q3: The reported performance gains are mostly incremental (although this is probably not unexpected with a relatively small change such as the weight parametrization on something as basic as feedforward neural networks with best practices for training them that have been established over the past decade).***
> > >
> > > We respectfully disagree with the reviewer and would like to emphasize that best practices for training feedforward neural networks for standard tasks such as classification do not necessarily transfer to training coordinate-based MLPs. Determining these best practices is a current and ongoing area of research. This line of work can be dated back to the finding of ``spectral bias'' [1,2], a phenomenon that hinders the performance of coordinate-based MLPs in approximating signals with high frequency components. To overcome this, Mildenhall *et. al.* [3] and  Tancik *et. al.* [4] developed positional encodings and random Fourier features that enable coordinate-based MLPs to fit high-frequency signals. Another example is SIREN [5], which employs MLPs with periodic activations to represent complex natural signals and their derivatives. Most recently, Müller *et. al.* [6] proposed multi-resolution hash encoding, speeding up the training of neural representations by several orders of magnitude. Moreover, training coordinate-based MLPs in the context of physics-informed neural networks (PINNs) presents unique challenges that are not resolved by traditional approaches for training feedforward neural networks [7,8,9].  We show that the proposed random weight factorization leads to significant performance gains for such scenarios (e.g. see the Computed Tomography and PINNs benchmarks) and is therefore an important contribution to training coordinate-based MLPs.
> > >
> > > ***Q4: The range of architectures is quite narrow, it seems like the parametrization would be straight-forward to plug into CNNs and Transformers if I'm not missing anything.***
> > >
> > > The reviewer is correct that the proposed parameterization can be easily plugged into other network architectures. While this is an important direction for future research, it is out of the scope of the current paper.  Here we would like to emphasize again that this work specifically aims to improve the performance of coordinate-based MLPs, which have recently been applied with great success across a diverse collection of tasks in computer vision, graphics and scientific computing [3,5,6,10].
> > >
> > >
> > > [1] Nasim Rahaman, Aristide Baratin, Devansh Arpit, Felix Draxler, Min Lin, Fred Hamprecht, Yoshua Bengio, and Aaron Courville. On the spectral bias of neural networks. In International Conference on Machine Learning, pages 5301–5310, 2019.
> > >
> > > [2] Ronen Basri, Meirav Galun, Amnon Geifman, David Jacobs, Yoni Kasten, and Shira Kritch- man. Frequency bias in neural networks for input of non-uniform density. arXiv preprint arXiv:2003.04560, 2020.
> > >
> > > [3] Ben Mildenhall, Pratul P Srinivasan, Matthew Tancik, Jonathan T Barron, Ravi Ramamoorthi, and Ren Ng. Nerf: Representing scenes as neural radiance fields for view synthesis. In European conference on computer vision, pages 405–421. Springer, 2020.
> > >
> > > [4] Matthew Tancik, Pratul Srinivasan, Ben Mildenhall, Sara Fridovich-Keil, Nithin Raghavan, Utkarsh Singhal, Ravi Ramamoorthi, Jonathan Barron, and Ren Ng. Fourier features let networks learn high frequency functions in low dimensional domains. Advances in Neural Information Processing Systems, 33:7537–7547, 2020.
> > >
> > > [5] Vincent Sitzmann, Julien Martel, Alexander Bergman, David Lindell, and Gordon Wetzstein. Implicit neural representations with periodic activation functions. Advances in Neural Information Processing Systems, 33:7462–7473, 2020.
> > >
> > > [6] Thomas Müller, Alex Evans, Christoph Schied, and Alexander Keller. Instant neural graphics primitives with a multiresolution hash encoding. arXiv preprint arXiv:2201.05989, 2022
> > >
> > > [7] Sifan Wang, Yujun Teng, and Paris Perdikaris. Understanding and mitigating gradient flow pathologies in physics-informed neural networks. SIAM Journal on Scientific Computing, 43(5):A3055–A3081, 2021.
> > >
> > > [8] Sifan Wang, Xinling Yu, and Paris Perdikaris. When and why PINNs fail to train: A neural tangent kernel perspective. Journal of Computational Physics, 449:110768, 2022
> > >
> > > [9] Aditi Krishnapriyan, Amir Gholami, Shandian Zhe, Robert Kirby, and Michael W Mahoney. Characterizing possible failure modes in physics-informed neural networks. Advances in Neural Information Processing Systems, 34:26548–26560, 2021
> > >
> > > [10] Maziar Raissi, Paris Perdikaris, and George E Karniadakis. Physics-informed neural networks: A deep learning framework for solving forward and inverse problems involving nonlinear partial differential equations. Journal of Computational Physics, 378:686–707, 2019.

---

> > > > ### Author Response · Authors · 2022-11-16
> > > > **Response to Reviewer C8fw (Part IV)**
> > > >
> > > > ***Q5: I don't find the argument around the proximity of parametrizations too convincing, I'd expect these to be increasingly far away from the origin, especially in high dimensions. This probably would make them hard to reach by gradient-based optimization, in particular in conjunction with L2 regularization (although I suppose it depends whether that is applied to the weight matrices or the actual parameters). I would appreciate some more discussion around this or some small experiment on an actual neural network architecture.***
> > > >
> > > > In Theorem 1, we rigorously prove that the distance between any network parameterizations is 0 in the factorized parameter space. This is also consistent with our geometric observation that each parameterization corresponds a hyperbola in the factorized parameter space and the distance between them become arbitrarily small for enough large values of the scale factors (see Figure 1). Motivated by this, we proposed the random weight factorization, which encourages initialization of the scale parameter away from the origin. This is accomplished by sampling the scale factors from a appropriate random distribution and re-parameterizing the weight matrices accordingly at initialization.
> > > >
> > > > To the best of our knowledge, L2 regularization is not widely used in building continuous neural representations via coordinate-based MLPs since it generally degrades their performance. To this end, we encourage the referee to check the official implementations of some representative examples for more details:
> > > >
> > > > -  DeepSDF: https://github.com/facebookresearch/DeepSDF
> > > >
> > > > - NeRF: https://github.com/bmild/nerf
> > > >
> > > > - SIREN: https://github.com/vsitzmann/siren
> > > >
> > > > - PINNs: https://github.com/maziarraissi/PINNs
> > > >
> > > >
> > > > ***Q6: I haven't seen any of the benchmarks before, so find it difficult to judge the reported performance gains.***
> > > >
> > > > The benchmarks presented here are standard for training continuous neural representations with coordinate-based MLPs. The tasks of image regression, shape representation, computed tomography and inverse rendering where originally proposed in [1] and have been widely used by the community ever since. In particular, *Dragon* and *Armadillo* are standard datasets for evaluating surface reconstruction algorithms in computer graphics  (http://graphics.stanford.edu/data/3Dscanrep), while *Lego* is a standard benchmark in neural rendering. For the task of solving PDEs, the advection equation is widely used in evaluating the performance of physics-informed neural networks (PINNs) [2], and the lid-driven-cavity problem is a standard benchmark in computational fluid dynamics [3]. In addition,  the examples of diffusion-reaction,  Darcy and Burgers equation are standard benchmarks in operator learning (see [4, 5] for more details).
> > > >
> > > > [1] Matthew Tancik, Pratul Srinivasan, Ben Mildenhall, Sara Fridovich-Keil, Nithin Raghavan, Utkarsh Singhal, Ravi Ramamoorthi, Jonathan Barron, and Ren Ng. Fourier features let networks learn high frequency functions in low dimensional domains. Advances in Neural Information Processing Systems, 33:7537–7547, 2020.
> > > >
> > > > [2] Aditi Krishnapriyan, Amir Gholami, Shandian Zhe, Robert Kirby, and Michael W Mahoney. Characterizing possible failure modes in physics-informed neural networks. Advances in Neural Information Processing Systems, 34:26548–26560, 2021
> > > >
> > > > [3] Charles-Henri Bruneau and Mazen Saad. The 2d lid-driven cavity problem revisited. Computers & fluids, 35(3):326–348, 2006.
> > > >
> > > > [4] Lu Lu, Pengzhan Jin, Guofei Pang, Zhongqiang Zhang, and George Em Karniadakis. Learning
> > > > nonlinear operators via DeepONet based on the universal approximation theorem of operators.
> > > > Nature Machine Intelligence, 3(3):218–229, 2021.
> > > >
> > > > [5] Zongyi Li, Nikola Kovachki, Kamyar Azizzadenesheli, Burigede Liu, Kaushik Bhattacharya, Andrew Stuart, and Anima Anandkumar. Fourier neural operator for parametric partial differential equations. arXiv preprint arXiv:2010.08895, 2020.

---

> > ### Comment · Reviewer_C8fw · 2022-12-09
> > **Thank you for the response**
> >
> > Thank you for providing results across a broader range of settings. I'm still hesitant to recommend acceptance, as I still do not see any attempt to optimize the learning-specific hyperparameters for a fixed setup. Nevertheless, a parametrization that works better under default settings may still be useful, so I will increase my score.

---

> > > ### Author Response · Authors · 2022-12-10
> > > **Response to Reviewer C8fw**
> > >
> > > Thank you for your comments and feedback. We address your follow-up comments here.
> > >
> > >
> > > As detailed in our previous response, we have performed comprehensive hyper-parameter sweeps that include learning-specific hyper-parameters such as learning rates. While it is unclear what the referees exactly means by *“I still do not see any attempt to optimize the learning-specific hyper-parameters for a fixed setup“*, here we provide an additional hyper-parameter study for the Advection equation benchmark in which we test all competing approaches across a range of learning-specific hyper-parameters, including initial learning rate step sizes, decay steps, decay rates, as well as random seeds. The range of hyper-parameters and results for this experiment are summarized in the following tables:
> > >
> > > |    Hyper-parameters   |        Values        |
> > > |:---------------------:|:--------------------:|
> > > |      Random seed      |         2,3,5        |
> > > | Initial learning rate | 0.001, 0.005, 0.0001 |
> > > |       Decay rate      |       0.9, 0.95      |
> > > |      Decay steps      |   2000, 5000, 10000  |
> > >
> > > |     Task     |           Metric          |    Case   |         Plain        |           AA          |          WN          |           RWF (ours)          |
> > > |:------------:|:-------------------------:|:---------:|:--------------------:|:---------------------:|:--------------------:|:-----------------------------:|
> > > | Solving PDEs | Rel. $L^2$ error ($\downarrow$) | Advection | 48.38\% $\pm$ 29.29\% | 48.76\% $\pm$ 30.19\% | 71.66\%  $\pm$ 7.33% | **31.03\% $\pm$ 29.80\%** |
> > >
> > > We remark that the larger test errors and  standard deviations observed here are because this benchmark is very sensitive to the learning rate and all methods fail to predict reasonable PDE solutions for very small initial learning rate $10^{-4}$. Nevertheless, it can be concluded that the proposed random weight factorization performs the best and is the most robust approach against different learning rate schedules.
> > >
> > > We would also like to emphasize that the results reported in Table 1 of the main manuscript do correspond to **optimized hyper-parameter settings** as reported in the literature for each benchmark and the approaches we are comparing against.
> > > Taken all together, we have already performed more than 10,000 numerical experiments that include extensive hyper-parameter sweeps, and all results we have reported thus far strongly support the main claims of our paper.
> > >
> > > We would be grateful if the referee could elaborate more on learning-specific hyper-parameters and what additional experiments he/she expects to see, and we will try our best to accommodate this request before the Dec. 12 deadline.

---

> > > > ### Author Response · Authors · 2022-12-11
> > > > **Discussion period ending soon**
> > > >
> > > > Dear Reviewer C8fw,
> > > >
> > > > We would like to thank the reviewer again for your time and effort.
> > > >
> > > > Given there is only ***1 days left*** in the discussion period, we wanted to double check if the reviewer had seen our latest comment and whether there were any final clarifications the reviewer would like. If the concerns of the reviewer are clarified and the reviewer is convinced of the novelty and contribution of our work, we sincerely appreciate it if the reviewer could update your review and score to reflect that.
> > > >
> > > > Once again, many thanks for your time and dedication to the review process, we are extremely grateful.
> > > >
> > > > Best,
> > > >
> > > > Authors

---

### Comment · Area_Chair_j3En · 2022-11-22
**Please respond as soon as possible if you still have questions on the paper.**

Please respond as soon as possible if you still have questions on the paper.

---

> ### Comment · Area_Chair_j3En · 2022-11-29
> **Please respond to the authors by Nov. 30**
>
> Please indicate whether the authors' rebuttal addresses your concerns.
>
> If you still have questions, please ask as soon as possible.

---

> > ### Comment · Area_Chair_j3En · 2022-12-05
> > **Zoom Meeting**
> >
> > For all reviewers, which have not responded to the authors, I will have to ask you to meet via Zoom. If you want to avoid such an additional step, please respond by Dec. 5.

---

### Decision · Program_Chairs · 2023-01-20

**Decision:**

Reject

**Justification For Why Not Higher Score:**

NA

**Justification For Why Not Lower Score:**

NA

**Metareview: Summary, Strengths And Weaknesses:**

The paper proposes a method for improving and accelerating the training of coordinate-based networks by proposing randomized weight factorization. This factorization generalizes weight normalization, and can be applied in the coordinate-based network setting for signal memorization.

The reviewers raised a major concern on experiments:

(1) The proposed method slightly outperformed baselines in their investigated applications. However, considering the provided sample standard deviations, all the p-values for the corresponding two sample t-tests are large. Therefore, the improvements are not statistically significant.

(2) The proposed method is not generic, and cannot be applied to other popular architectures in as natural language processing and computer vision.

**Summary Of Ac-Reviewer Meeting:**

NA